# A continuum model of transcriptional bursting

**Adam M Corrigan[1,2], Edward Tunnacliffe[1,2], Danielle Cannon[1,2], Jonathan R Chubb[1,2]***

[1]Division of Cell and Developmental Biology, University College London, London, United Kingdom; [2]Laboratory for Molecular Cell Biology, University College London, London, United Kingdom

**Abstract** Transcription occurs in stochastic bursts. Early models based upon RNA hybridisation studies suggest bursting dynamics arise from alternating inactive and permissive states. Here we investigate bursting mechanism in live cells by quantitative imaging of actin gene transcription, combined with molecular genetics, stochastic simulation and probabilistic modelling. In contrast to early models, our data indicate a continuum of transcriptional states, with a slowly fluctuating initiation rate converting the gene between different levels of activity, interspersed with extended periods of inactivity. We place an upper limit of 40 s on the lifetime of fluctuations in elongation rate, with initiation rate variations persisting an order of magnitude longer. TATA mutations reduce the accessibility of high activity states, leaving the lifetime of on- and off-states unchanged. A continuum or spectrum of gene states potentially enables a wide dynamic range for cell responses to stimuli.

## Introduction

Transcription of genes is discontinuous, occurring in irregular bursts or pulses of activity, interspersed by irregular intervals of inactivity (*Golding et al., 2005*; *Chubb et al., 2006*; *Raj et al., 2006*). Bursting transcription is conserved in all forms of life, from prokaryotes (*Chong et al., 2014*) to mammalian cells and tissues (*Suter et al., 2011*; *Bahar Halpern et al., 2015*; *Harper et al., 2010*). The irregular nature of transcriptional bursting is proposed to be a major driver of spontaneous heterogeneity in gene expression, which in turn drives diversity of cell behaviour in differentiation and disease (*Raj and van Oudenaarden, 2008*; *Eldar and Elowitz, 2010*). Bursting reflects the underlying mechanisms of transcriptional regulation, and measures of bursting can reveal the dynamic processes absent from standard population average measures of RNA expression.

The standard framework used to describe transcriptional fluctuations compares one state and two state models (*Raj and van Oudenaarden, 2008*). In the one state model, transcription occurs with a constant probability, which for moderately and strongly transcribed genes, will generate a low variance in their total transcribed RNA per cell. In some contexts, notably budding yeast (*Zenklusen et al., 2008*), the variance in RNA abundance measured by single molecule RNA fluorescence in situ hybridisation (smFISH) (*Femino et al., 1998*; *Mueller et al., 2013*) can fit this one state scenario, where the distribution of RNA per cell is well characterised by a Poisson distribution. In many other contexts, the one state model does not fit the smFISH data, with measured RNA abundance showing too much variability between cells than can be produced by a constantly active gene. To explain this increased variance, the more complex random telegraph (or two state) model is often invoked (*Paulsson, 2005*). In this model, the gene switches stochastically between an active state, where mRNA production occurs with constant probability per unit time, and an inactive state, with no mRNA production. The extra state increases the potential variability in output from cells, and can

**\*For correspondence:** j.chubb@ucl.ac.uk

**Competing interests:** The authors declare that no competing interests exist.

**eLife digest** Understanding how gene activity is regulated relies on accurate measurements of the output of genes. Proteins are generated from genes via a multi-step process. In the first step, called transcription, the DNA of a gene is copied by complex cell machinery to create molecules of mRNA. Subsequently, these mRNA molecules are 'translated' into proteins.

Previous studies have assayed gene transcription by measuring mRNA production in millions of cells at the same time. The resulting measurements give the impression that transcription occurs as a continuous, smooth process. However, when individual gene transcription is measured in single cells, mRNA production between cells is unexpectedly variable. This challenged the view that transcription is a continuous process.

One idea that explains this variability – the "two-state" or "bursting" model – proposes that genes switch between "on" and "off" states with a certain probability. Thus, at any one time, a gene will be off in many cells and on in others. However, the methods used in these experiments measure mRNA in dead cells, and so the dynamic switching of genes between on and off states was presumed, but not accurately measured.

Corrigan et al. have now imaged the transcription of a single gene – a gene for a protein called actin – in living cells of an amoeba called *Dictyostelium*. Genetic techniques and computational modeling were then used to explore what affects the variability in this gene's activity. These approaches revealed that transcription occurs across a spectrum of activity, rather than in rigid on or off states. The transcription process itself may also contribute to where a gene's activity sits on this spectrum. Furthermore, Corrigan et al. found that a specific DNA sequence found at the start of the actin gene, that is also found in many genes in complex life-forms, is required for the gene to reach the highest levels of activity on the spectrum.

This spectrum of activity states could allow cells to finely tune their responses to the signals they receive. A future challenge will be to assess how the activity of other genes compare to the actin gene and to discover what underlies the variation in the timing of transcription's different stages.

therefore predict the observed extra spread in transcript abundance in the cell population (*Singer et al., 2014*).

Use of the two-state model in fitting smFISH and protein distributions allows estimates of the parameters of the transcriptional fluctuations, usually the burst size (number of transcripts produced in a burst) and burst frequency (the frequency with which a burst occurs) (*Carey et al., 2013*; *Dar et al., 2012*). However, these dynamic properties are usually inferred from a population distribution at a single time point, assuming each cell is part of a homogeneous population with fixed values of the switching rates, transcript production rate and transcript lifetime. In other words, the perception has emerged that transcriptional bursting is a product of molecular noise, rather than a process responsive to the demands of the cell. A rethink is required, not least because of recent work demonstrating burst size and frequency are quantities that can be modulated by extracellular signals (*Molina et al., 2013*; *Corrigan and Chubb, 2014*; *Senecal et al., 2014*) and cell properties such as volume and cell cycle stage (*Padovan-Merhar et al., 2015*; *Muramoto et al., 2010*). These studies challenge the notion, central to the standard two state model, that a population of cells consists of those where the gene of interest is 'off' and those where the gene is 'on' with a constant probability of firing.

To make accurate models of transcriptional fluctuations and how they are regulated, it is critical to directly observe and quantify how transcription evolves over time. To directly measure features such as burst size and burst frequency requires data capture of complete sequences of bursts, rather than snapshots. Imaging transcriptional output in living cells is possible using RNA detection systems based upon the binding of a bacteriophage coat protein to stem loops of RNA (*Chubb et al., 2006*; *Bertrand et al., 1998*). An array of sequence encoding stem loops, such as MS2 or PP7, is inserted after the promoter of a gene of interest. When the gene is transcribed, the nascent RNA stem loops are detected with a fluorescent MS2 (MCP) or PP7 (PCP) coat protein, which is constitutively co-expressed. The system is visualized using time-lapse fluorescence microscopy, with a nuclear spot

indicating active transcription of the gene of interest. The intensity of the spot reflects the instantaneous nascent RNA load at the gene and changes in spot intensity reflect how the RNA load fluctuates over time. These tools allow direct observation of the dynamics of transcription regulation, revealing insights into the mechanics of Poissonian transcription (*Larson et al., 2011*) and developmental regulation (*Corrigan and Chubb, 2014*; *Muramoto et al., 2012*; *Garcia et al., 2013*; *Stevense et al., 2010*; *Bothma et al., 2014*; *Lucas et al., 2013*). Previous work has interpreted the appearance and disappearance of transcription spots in terms of 'bursty' transcription (*Muramoto et al., 2010*; *Masaki et al., 2013*). Here, the exponential nature of the 'ON' and 'OFF' time durations of the transcript spot was related to the two state model, with the exponential behaviour proposed to reflect rate-limiting steps in transitions between the active and inactive states. Although this is an appealing inference, how spot fluctuations actually reflect the dynamics of the transcription machinery is unclear.

In this paper, we test the current models for explaining transcriptional fluctuations, using a combination of live cell imaging, computational modelling and simulation, and targeted mutations of gene and promoter structure. We use a probabilistic approach to infer dynamics at the molecular level from fluctuations in spot intensity. We make quantitative measurements of the transcription site RNA abundance and the retention time of nascent RNA at the gene. We use these measurements to train candidate hidden Markov models to describe the underlying initiation of RNA polymerases, and find that a spectrum or continuum of initiation rates describes experimental data more accurately than a binary off/on model or discrete levels of activity. Finally, we investigate how the processes of transcription elongation and initiation contribute to the transitions of the gene over this spectrum of activity states.

## Results

### Measurement of transcription fluctuations

To monitor transcriptional dynamics in living cells, we integrated an array of MS2 stem loops after the promoter of the endogenous *actin5* locus, a strongly expressed actin gene in undifferentiated *Dictyostelium* cells. Transcription continues after the MS2 loops into the coding sequence, then native terminator, to generate a full length transcript of around 2.5 kb. We visualized the resulting transcription dynamics using time-lapse fluorescence microscopy and extracted time series of spot intensities using custom-built software integrating both cell tracking and spot detection (*Corrigan and Chubb, 2014*). The movie sequence in *Figure 1A* illustrates the tracking of a typical cell, showing the fluctuations in spot intensity over time. *Figure 1B* shows the measured transcription spot intensity for the cell in *Figure 1A*, with a kymograph of the spot fluctuations. For most genes studied, the durations for which a spot is present or absent are often measured to have approximately exponential distributions, which is the case for *act5* (*Figure 1—figure supplement 1*) (*Muramoto et al., 2012*). Exponential timescales have been inferred to represent modulation of gene activity, between the ON and OFF states of the two-state model, with a rate-limiting step determining switching between states (*Golding and Cox, 2006*). A more simple possibility is that stochastic fluctuations of a gene with no OFF state - the one state model - could give rise to pulses and intervals between pulses with the experimentally observed lifetimes.

### Pulsing theory

To test if a one state model, where the probability of a polymerase initiating is constant over time, can give rise to the distributions of spot durations observed experimentally, we constructed a Monte-Carlo simulation framework describing the MS2 system, incorporating polymerase initiation (the rate at which polymerases begin transcription) and elongation (the velocity of the polymerase while transcribing) (*Figure 1C,D*). Additional features, such as the number of gene states, and termination/release rate (the rate at which the RNA leaves the gene) were omitted at this stage. By defining a signal detection threshold above which a spot is determined to be present, we calculated the pulse and interval length distributions from the simulated data. The average pulse duration depends strongly on the initiation rate, but was also substantially sensitive to the detection threshold and frame interval specified in the simulations. The pulse duration distribution had exponential distributions at long timescales, with deviations at very short timescales (*Figure 1E*). This was surprising, as

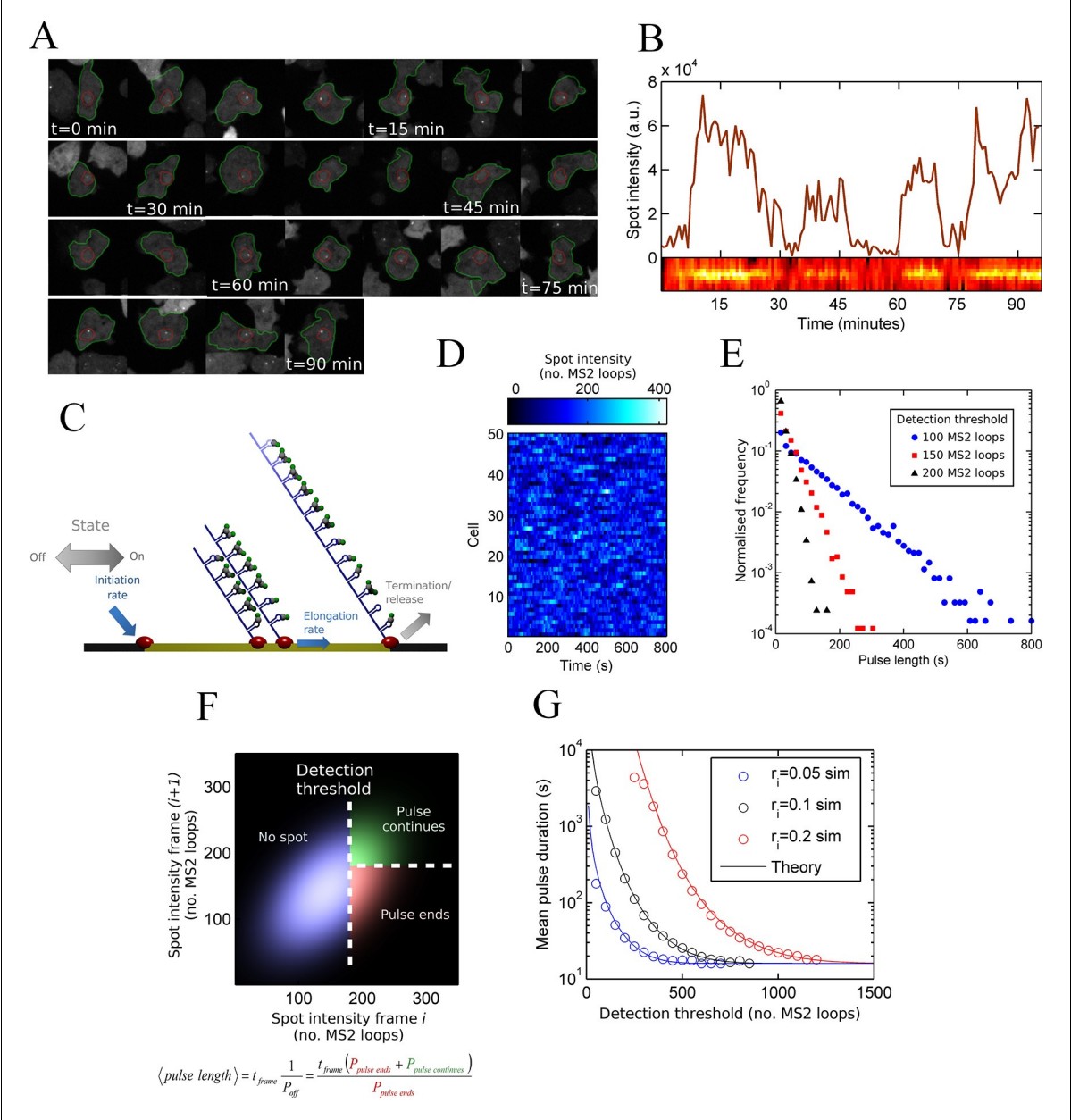

**Figure 1.** Measurement and theory of transcriptional fluctuations See also *Figure 1—figure supplements 1* and *2*. (A) Montage of a cell identified and tracked throughout a time lapse movie showing the transcription spot fluctuating over time. Detected cell (green) and nuclear (red) boundaries are shown. (B) (Upper) Spot intensity trace for the cell shown in A. (Lower) Kymograph extracted from image, aligned with time axis of upper graph, showing the fluctuations in intensity of the region around the spot. (C) Monte Carlo simulation of MS2 system. Binding of polymerases at the start of the gene (initiation) and single nucleotide elongation steps are modelled as processes with one rate-limiting step. Additional steps could be added, such as termination/release from the gene. To simulate systems with switches in initiation rate, single rate-limiting steps are used to transition between different initiation states. (D) Simulated transcription site intensity fluctuations (total number of stem loops) for a promoter with a constant Poisson initiation rate. (E) Histogram of pulse durations for different detection thresholds. A pulse is defined as successive frames where the transcription site intensity is above a threshold number of loops. Experimentally, the threshold of detection is the intensity at which a spot is identifiable over background noise, and depends on the imaging conditions. (F) Two-dimensional histogram calculated from the bivariate Gaussian theory, showing the probability distribution of the transcription site intensity in two successive frames. Blue region - spot intensity below threshold in current frame; green region - intensity above threshold in both current and next frames; red region - spot intensity above threshold in current frame but below threshold in next frame. The average pulse duration is determined from the probability of the transcription spot disappearing between one frame and the next: P (off) = P(red)/(P(green) + P(red)). (G) The bivariate Gaussian theory accurately predicts the pulse durations of simulated data. Comparison of theory and

*Figure 1 continued on next page*

*Figure 1 continued*

simulation are shown for three different initiation rates ($r_i$). Therefore, the duration of a visible transcription pulse depends on properties such as the exposure time, detection sensitivity and frame interval, and does not provide a simple readout of gene activity fluctuations.

The following figure supplements are available for figure 1:

**Figure supplement 1.** Experimental pulse durations obtained by applying various thresholds of detection: low - 4000 arbitrary intensity units (a.u.), middle - 8000 a.u. and high 16,000 a.u.

**Figure supplement 2.** Agreement between simulations and bivariate Gaussian theory of spot frequency (fraction) (right) as a function of detection threshold.

the exponential distribution was previously inferred to support the two-state model. Here, our simulations clearly show such a distribution can arise from a more simple one-state scenario.

An exponential distribution of pulse durations has one parameter, a characteristic timescale, and arises from there being a constant probability of the pulse ending at any given time. This probability is independent of the duration for which the pulse has been sustained. It is tempting to infer from this probability the rate of switching from an active to an inactive state of a two state gene. However, with exponential behaviour observed for our simulation of constant polymerase initiation rates (this is only one state), it is necessary to consider additional contributions to the likelihood of a spot disappearing from one imaging frame to the next.

To investigate the origins of exponential behaviour, we constructed a theory of spot appearance and disappearance for the live MS2 system. By considering the rate of initiation of polymerases at the start of the gene and the number of RNA stem loops produced by a polymerase as it moves along the gene, we calculate the joint probability distribution linking the spot intensity in the previous frame with the current frame. (*Figure 1F*, full details in the appendix). The theory successfully predicts the characteristic exponential timescale of pulse duration in terms of the one state model, incorporating the polymerase elongation rate, the frame interval and threshold of spot detection. In simple terms, the pulse duration reflects the balance between polymerase initiation and elongation or termination - in order for a spot to remain present in the next frame, a sufficient number of new polymerases must be initiated to replace the loss of spot intensity from cleaved RNA leaving the transcription site. Intuitively, high initiation rates are likely to give rise to large numbers of initiations per frame, giving long pulses. Increasing the threshold of spot detection decreases the likelihood of sufficient initiations and thus decreases the pulse duration. These predictions are in agreement with the Monte Carlo simulations of the MS2 system (*Figure 1G* and *Figure 1—figure supplement 2*). The initiation and elongation rates have approximately inverse effects, with initiation scaling directly with spot intensity (as expected intuitively) and elongation scaling inversely. The effect of slow elongation increasing spot intensity can be thought of in terms of a polymerase 'traffic jam', with several polymerases building up behind a slower one, causing a build up of nascent RNA on the gene (*Darzacq et al., 2007*). Additionally, the frame interval of observation influences the effective pulse duration, as longer frame intervals increase the likelihood of a gap between pulses being missed between frames. Overall, this theory shows that the length of time for which a spot is visible is not simply related to switches between proposed gene states in two state models and explains the dependence of the pulse duration on the imaging signal-to-noise (*Muramoto et al., 2010*) and frame interval (*Masaki et al., 2013*).

Since the theory shows that the simple presence of a spot does not provide insight into what activity state the gene is currently in, we must make precise measurements of transcription site intensity in order to detect initiation rate changes (changes in gene state). As depicted by the cartoon in *Figure 2A*, a fluctuation in intensity might be consistent with either the one state (Poisson, top) or multi-state (bottom) models. Without calibration of the intensity fluctuations in terms of numbers of RNAs, it is difficult to discount either model. One strategy that can be used is an autocorrelation analysis, which measures the magnitude and timescale of intensity fluctuations (*Larson et al., 2011*). The autocorrelation measures the ratio of intensity fluctuations relative to the mean intensity. For a Poisson (one state) gene, where initiations behave independently of the previous or subsequent polymerase, this ratio is related to the average number of bound polymerases, allowing the initiation

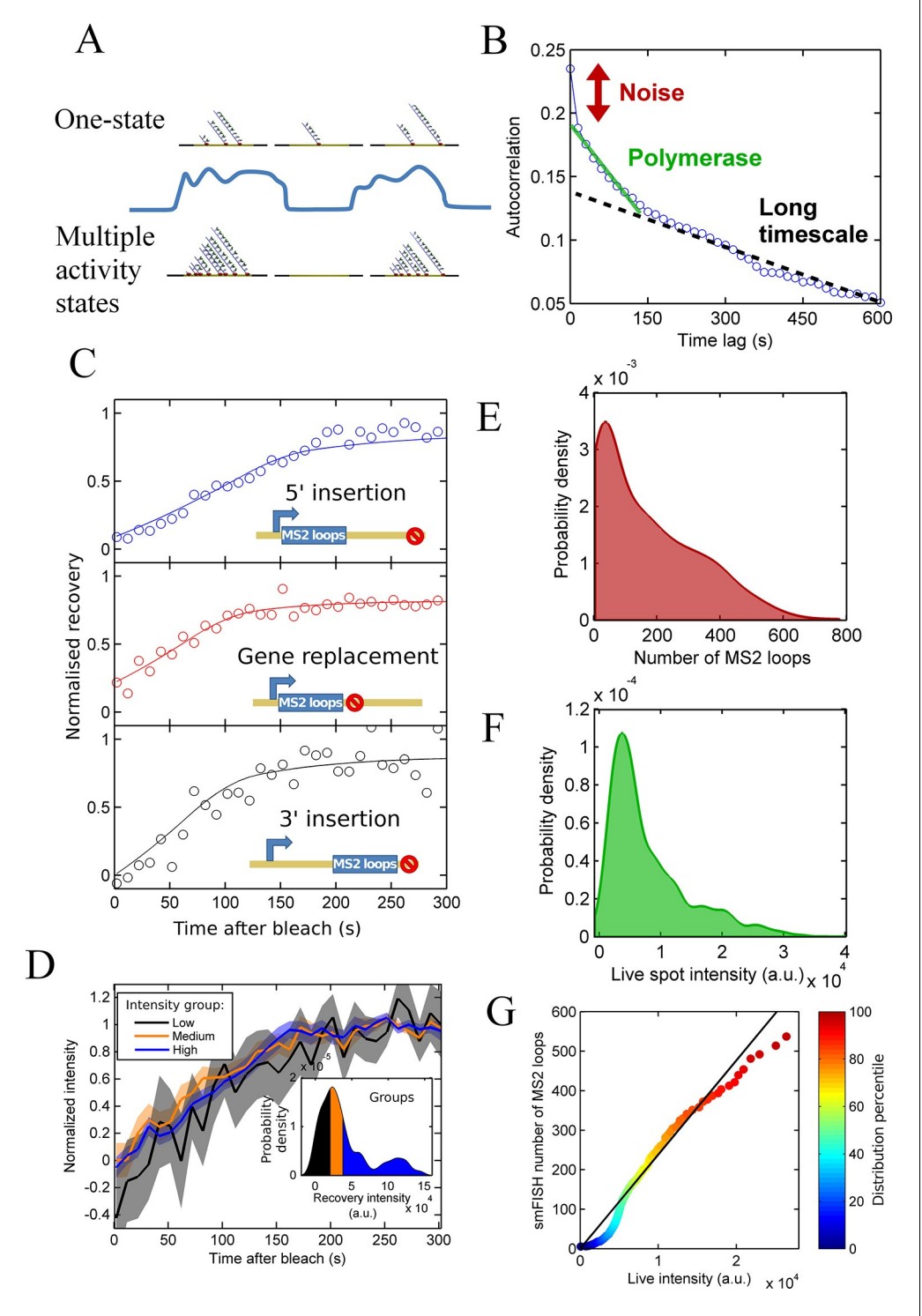

**Figure 2.** Calibration of MS2 system provides quantitative detail of polymerases at the transcription site. See also *Figure 2—figure supplements 1* and *2*. (A) The correspondence between spot intensity and number of MS2 loops at the transcription site strongly influences the type of model which accurately describes the experimental data. Depending on the actual detection threshold, the blue intensity trace could be generates by either the one state (top) or multiple activity state scenarios (bottom). (B) Autocorrelation of transcription spot traces. The autocorrelation can be decomposed into three parts: measurement error (noise), polymerase contribution, and longer timescale fluctuations. Classification and distinction between the three parts is discussed in detail in the text. (C) FRAP curves showing recovery of TS intensity after photobleaching for different configurations of MS2

*Figure 2 continued on next page*

*Figure 2 continued*

loop position. The inset cartoons illustrate the arrangement of loops after the *actin5* promoter. Solid line shows best fit to model described in the text. For the 5' MS2 loop insertion, n=30 cells, for the 3' loop insertion, n=32 and for the gene replacement loop insertion, n=25, with each insertion line analysed on 4+ experimental days. (D) Grouping of FRAP curves based on the recovery intensity, showing no clear variability in dwell time as a function of intensity. The 5' MS2 insertion cell line was used here, with data from 56 cells (captured on 5+ experimental days) divided into 3 groups for high, medium and low spot intensity (inset). The experimental variability is shown as standard error. (E) Intensity distribution of transcription spots measured by smFISH using a probe hybridising to the inter loop region of the MS2 loop array. Plot shows the probability density function. The intensity of one MS2 RNA is calculated from cytoplasmic spots, and used to calibrate the nascent FISH transcription spot intensity in terms of the number of complete MS2 RNA molecules each consisting of 24 loops. For calibration, an average of 53,150 cytoplasmic RNA spots were used to measure single molecule fluorescence. 594 transcription spots were measured using smFISH. (F) Intensity distribution of transcription spots measured in live cells using MCP-GFP fluorescence. 1449 transcription spots were measured. (G) Calibration of MS2 live TS intensity using smFISH measurements. Comparing percentiles of the smFISH (E) and live distributions (F), allows the live TS intensity to be interpreted in terms of the number of stem loops present. The colour of the points indicates the percentile of the distribution.

The following figure supplements are available for figure 2:

**Figure supplement 1.** Experimental spot data is not consistent with a model of constant activity.

**Figure supplement 2.** Experimental spot data is not consistent with a binary on/off model (two-state) of transcription initiation.

rate to be estimated. This was previously used to infer the initiation and elongation rates from a *POL1* promoter inserted upstream of the *GLT1* gene in budding yeast (*Larson et al., 2011*), specifying the time for the correlation to decay to zero as the time for which a single polymerase contributes to the spot intensity.

Applying a similar analysis to transcription of the *Dictyostelium act5* gene (*Figure 2B*), we find that inference of polymerase properties is complicated by an additional long-timescale decay of several minutes to tens of minutes (black dotted line). The additional slow decay means the RNA level at the gene is correlated beyond the dwell time at the gene of a single RNA. This argues against a Poisson model and indicates a model with two or more transcription states would better describe *act5* transcription. If the autocorrelation is recalculated using only frames where a spot is present, the slow decay remains, suggesting that the active periods where a spot is present are not defined by a single constant initiation rate; in other words there are multiple 'ON' states. By subtracting a linear estimate of the long timescale decay from the autocorrelation, we calculated estimates of the RNA dwell time and polymerase load on the gene (*Larson et al., 2011*). The point where the autocorrelation deviates from the long decay gives an approximation of the dwell time as 190 ± 20 s. The average number of polymerases contributing to the intensity is estimated from the magnitude of the deviation at zero lag time as 20–27 polymerases. The cells are predominantly in G2 (*Muramoto and Chubb, 2008*), so these autocorrelation-based estimates are for the polymerase load across 2 alleles of the *act5* gene comprising the transcription spot.

## Calibrating RNA load and dwell time

As described above, the observed pulsing properties also depend on the RNA dwell time- the time the RNA spends at the transcription site during elongation and termination before it is released from the gene. As an independent measure of dwell time, we used fluorescence recovery after photobleaching (FRAP) measurements on MS2 spots (*Muramoto et al., 2012*). The MCP-RNA interaction is stable (*Maiuri et al., 2011*), so recovery of the spot intensity after bleaching is determined by transcription of new MS2 loops. In the absence of photobleaching, spot intensities fluctuate over time, therefore a bleached spot is not expected to recover to its pre-bleach intensity. We therefore normalized the curves based on the recovered intensity. Recovery of a single endogenous gene is highly stochastic, as expected from simulations of photobleaching and recovery (see below). Therefore, we averaged over 20–30 cells in order to extract the typical RNA dwell time. The average

recovery curve was fit to a processive polymerase model of the intensity recovery to estimate the elongation rate and termination time (full details in the appendix). We calculated the dwell time for three different gene constructs, with the MS2 loops integrated into the 5' gene region, the 3' region, and replacing the entire *actin5* coding region (*Figure 2C*). The dwell time for the 5' insertion was longer than for the other insertions, as expected as the polymerase will continue to transcribe after the MS2 array, whereas with the other two cell lines, termination will occur immediately upon completion of synthesis of the array. Using the difference in dwell times and assuming a constant rate of elongation across the gene, we calculate the elongation rate as 22 nt.s$^{-1}$ and termination time as 60-70 s. This gives a total dwell time of around 170 s, for the 5' MS2 insertion, roughly in agreement with the autocorrelation-derived approximation on the same allele.

As shown by our pulsing theory, spot intensity variations may reflect either fluctuations in the initiation rate or dwell time. If the dwell time is systematically different for spots of different intensity, we would observe this by studying the recovery times of high and low intensity spots. We divided the FRAP data into three equal groups based on their recovery intensity (inset of *Figure 2D*) and calculated the average recovery curve for each group. *Figure 2D* shows that no significant differences in recovery time are observed for the different intensity groups, suggesting dwell time variations are not the primary source of the spot intensity variability.

We then calibrated live RNA spot intensity in terms of the number of MS2 stem loops by comparison with smFISH using a probe for the MS2 RNA. We used FISH-quant (*Mueller et al., 2013*) to estimate the intensity of one mature cytoplasmic RNA then used this information to calibrate the number of nascent MS2 loops at the transcription site (*Figure 2E*). We compared the distribution of these nascent RNA counts with the intensity distribution of transcription site MCP-GFP intensity measurements from live cells immediately prior to fixation for smFISH (*Figure 2F*). The two distributions are aligned by calculating the percentiles of each distribution and using these values as calibration points (*Figure 2G*). Due to the differing precision with which the intensity can be calculated in live and fixed samples, there is some deviation from a linear relationship at the extreme ends of the calibration curve, and a kink at low intensities where the detection threshold of live spots is higher than in fixed spots, however the overall trend can be used to calibrate the spot intensities measured in live cells in terms of the number of MS2 stem loops present.

After calibrating the system, we used our pulsing theory outlined above to estimate the initiation rate. We measured how the mean pulse duration changes as the detection threshold is varied. For a one-state (constant activity) gene, the pulse duration would lie on a contour calculated using our theory. *Figure 2—figure supplement 1* shows that the experimental data does not lie on a single contour of constant initiation rate, consistent with the autocorrelation results in *Figure 2B*. Similarly, contours calculated from a simple two-state implementation, with on- and off-rates chosen to produce the experimentally observed spot frequency (details in the appendix), cannot match the variation of pulse duration and spot frequency as the detection threshold is varied (*Figure 2—figure supplement 2*). Instead, the behaviour at high thresholds suggests a short-lived higher initiation rate.

*Dictyostelium* cells are almost exclusively in G2, owing to a very short S phase and complete absence of G1 (*Muramoto and Chubb, 2008*). Therefore almost all cells will have replicated the MS2-tagged gene and have two genes, held together by sister chromatid cohesion, contributing to a single resolved spot. Transcription of the unreplicated gene will not confuse analysis, as *act5* transcription is not robustly detected in the first 15 min after mitosis (*Muramoto et al., 2010*), when euchromatin is replicated (*Muramoto and Chubb, 2008*). A higher initiation rate state is possible in the situation where both copies are simultaneously active - giving a potential third state with double the initiation rate of a single copy.

## Hidden Markov Modelling of Initiation Rate Modulation

To test the possibility of this third state, we applied probabilistic modelling to our quantitative data to assess the extent to which three state or higher models can describe the observed transcription dynamics. Since a two-state model of transcriptional activity, with an 'off' state and an 'on' state of constant initiation rate, does not describe the live data adequately, we asked whether an improvement could be made by including additional active states with different initiation rates. It is not possible to directly observe polymerase initiations using the MS2 system, so we used hidden Markov

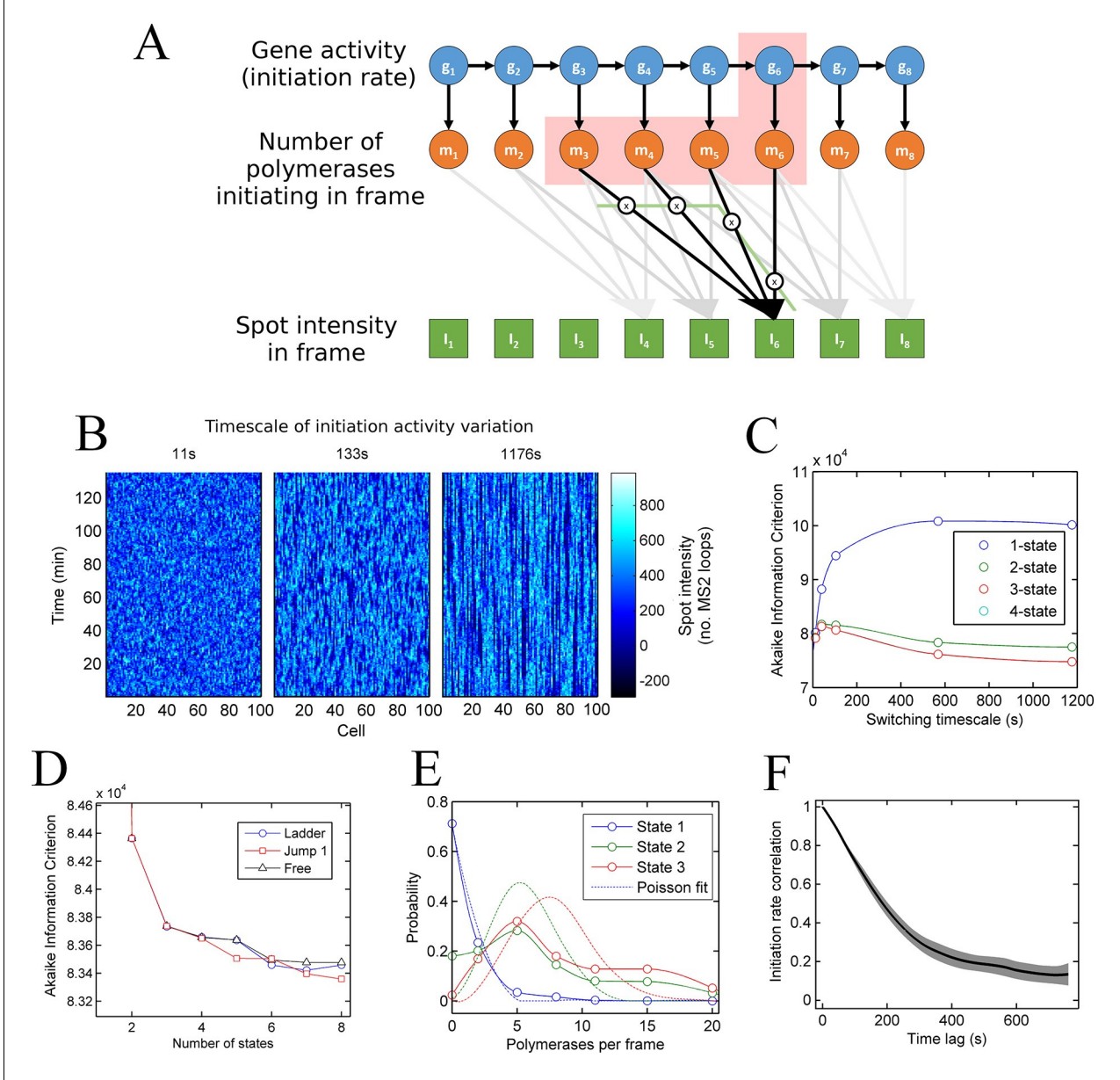

**Figure 3.** A continuum of transcriptional states. See also *Figure 3—figure supplements 1* and *2*. (**A**) Architecture of a hidden Markov model (HMM) to describe transcription spot intensity in the case where polymerases remain at the transcription site for up to 4 frames. The hidden state at a given point in time consists of the gene-state at the current time ($g_t$) and the number of polymerases (m) which have been initiated in the previous 4 frames [$g,m_i, m_{ii},m_{iii},m_{iv}$], highlighted by the red background. With approximately processive polymerase behaviour, polymerases initiated in the current frame will be near the start of the gene and thus have transcribed few MS2 loops; polymerases initiated in previous frames have transcribed more MS2 loops by the current frame. The polymerase states, weighted by the expected number of loops per polymerase (x), combine with the measurement error to give the observed state $I_t$ (green). (**B**) Simulated transcriptional fluctuations based on a 3-state model, with three panels corresponding to different timescales of switching between transcriptional states. The right panel (timescale of variation 1176 s) has longer pulses- reflecting the slower switching between initiation rate states. (**C**) Testing the HMM framework on the 3 state simulation from B. As described in the text, the AIC (Akaike's Information Criterion) is reduced for optimal models, while penalizing overly complex models via the number of free parameters. The one state fit has the highest value of AIC, regardless of the switching timescale. The 2-state fit does much better and the 3-state fit better still, with a reduced AIC. A 4-state fit gives no additional improvement over the 3-state fit and is hidden by the 3-state curve. (**D**) Increasing the number of possible initiation rate states improves the likelihood that the model reflects the experimental transcription data. AIC for models of increasing numbers of initiation states. While 1- and 2-state models do not adequately describe the data, the quality of the fit continues to significantly increase as the number of states increases from 3 upwards. The three curves indicate different rules for allowed transitions between states- 'ladder' means the gene can move up or down one state per time, 'jump 1' allows a change of up to 2 states and 'free' is unconstrained switching of the gene between states. These data represent a typical experiment, with data from 145 different cell tracks comprising 6350 individual time points. Three further 3 independent biological replicates gave similar

*Figure 3 continued on next page*

*Figure 3 continued*

conclusions. A decrease in AIC of 10 (note: the vertical axis units are scaled by $10^4$) is significant at the 1% level (p=0.007). A more extensive treatment of the statistics is included in the Supplementary Material. (E) Probability distribution of the number of polymerases initiated per frame for each state of a three-state model, calculated using a modified forward-backward algorithm. Attempted Poisson fits for each state are shown by the dotted lines. The distributions were strikingly non-Poissonian, with $\chi^2$ = 5059 (p=0) and 3152 (p=0) for states 2 and 3. For state 1, $\chi^2$ =10.24, but we cannot reject $H_o$ because of no degrees of freedom. Data from a representative biological replicate are shown. (F) The timescale of initiation rate fluctuations revealed by autocorrelation analysis. The curve shows the decay in the correlation as a function of time, with the initiation rate largely uncorrelated with the rate 5–6 min before.

The following figure supplements are available for figure 3:

**Figure supplement 1.** Accuracy of measurement of state initiation rates and state switching rate using hidden Markov model methods (Baum-Welch algorithm).

**Figure supplement 2.** Cumulative number of polymerases initiated as a function of time, calculated using a custom Gibbs sampling method.

modelling to infer the likelihoods of different numbers of active states from the measured live cell intensity data.

In simple terms, a hidden Markov model (HMM) describes a sequence of observations in terms of a sequence of underlying, 'hidden' states. By extension from the two-state model, we define a 'state' as a configuration with a single constant (Poisson) initiation rate. In terms of transcription dynamics, the hidden state corresponds to the underlying initiation rate of the system. This rate cannot be observed directly and instead must be inferred from measurements of the transcription site intensity, which is related probabilistically to the initiation rate. Using standard HMM techniques, the magnitudes and dynamics of the underlying initiation rate can be optimized to maximize the probability of generating the experimental data from the model. For the case of the MS2 system, the spot intensity does not depend only on the instantaneous initiation rate, because polymerases initiated in previous frames are still present on the gene and also contribute to the spot intensity. To take account of these polymerase contributions, we constructed a model (full details in the appendix, architecture depicted in *Figure 3A*) with hidden states representing the initiation rate (blue circles) and the number of polymerases initiated (orange circles), inferred from the sequence of transcription site intensities (green squares). A benefit of our two-layer HMM framework is that we explicity model the polymerases uncoupled from transitions between states of gene activity. This means that the distribution of polymerases initiated in each state can be estimated (see below).

To test how well models with discrete states of activity describe the data, we calculate the probability of obtaining the experimental data using the model parameters. This probability is known as the likelihood. We use a standard approach to compare models of different complexities, by calculating the Akaike Information Criterion (AIC) from the likelihood (*Singer et al., 2014*). The AIC is used to find the best approximation in situations where the real scenario is likely to be highly complex. Models are penalized by their number of free parameters, preventing overfitting by excessively complex models. The AIC estimates the prediction error of the candidate model, giving a lower numerical value for models which more accurately describe the experimental data.

To test the validity of the use of the HMMs and associated algorithms, we tested the framework on simulated data, with known numbers of promoter states, and known timescales of switching between states (*Figure 3B*). In these cases, the 'hidden' initiation states and numbers of initiated polymerases are known, allowing the accuracy of the probabilistic methods to be measured. Applying the hidden Markov approach to a simulated 3 state model accurately estimated 3 discrete initiation states as the point at which AIC is minimized (*Figure 3C*). 1 and 2 states give a higher AIC (lower likelihood), whilst 4 shows no improvement over 3 states. In addition, this approach robustly measured the initiation rate of each state, as well as the timescale of switching between the activity states (*Figure 3—figure supplement 1*).

We next applied the model fitting to experimental data for transcription of the *actin5* gene. We carried out 4 independent time lapse experiments, with multiple fields of view in each experiment, capturing data on 44–145 cells per experiment, giving 1686–6350 individual cell-frames, per experiment. *Figure 3D* shows the AIC for models with discrete numbers of initiation rate states. The 2-

state model shows a strong reduction in AIC compared to a 1-state model, with the 3-state model better still. The curve does not plateau after the 3 state model, instead, the fit to the data is continually and significantly improved by adding further initiation states. The continually improving fit does not depend on the permitted mechanisms of switching allowed from state to state (*Figure 3D*). This continual improvement in fit occurs if the gene constrained to switching between states one step at a time (*Figure 3D*- 'ladder'), in steps of up to two states ('Jump 1'), or is unconstrained ('Free'). The most realistic interpretation of the continually improving fit with more states is that the gene switches in activity over a spectrum or continuum of states, rather than a small number of discrete activity states.

In *Figure 3D* the largest improvement in fitting occurs up to 3 states; therefore, is a model with three initiation rate states appropriate for the *act5* gene? Although the slow rate of improvement in fit for higher numbers of states may suggest three states are adequate, closer analysis of the optimal 3-state fit reveals its inability to capture the full variability of the transcription site intensities. The values of the estimated initiation rates are not uniformly spaced (as would be expected for two alleles undergoing random telegraph transcription), but have a low state and a very high state in addition to an off-state. Furthermore, *Figure 3E* shows strongly non-Poisson distributions for the predicted number of polymerases initiated per frame in each state of the three state model, inconsistent with a constant initiation rate within each state. In finding the best 3 state fit, the modelling process effectively forced outlying data into non-Poissonian states. The wide polymerase distributions imply either that the initiation rate is varying within a state, or that transitions are occurring between states on a timescale significantly faster than the frame interval.

To determine the dynamics of the gene state and number of initiated polymerases, we used a Gibbs sampling algorithm (see appendix). The estimated number of polymerases initiated in each frame is approximately independent of the number of states chosen for the model. The initiation rate, the number of events per unit of time, is difficult to define as an instantaneous measurement; instead, by plotting the cumulative number of polymerase initiations as a function of time the initiation rate can be taken from the gradient. For a typical cell in the experimental data, the cumulative polymerase plot is composed of linear segments (*Figure 3—figure supplement 2*) representing periods of time over which the initiation rate is constant. We measured the initiation rate using an edge-preserving smoothing filter and found a diversity of such gradients, indicating a spectrum of initiation rates, again implying the transcriptional behaviour is not adequately described by a few discrete levels of activity. The most realistic model to account for this additional complexity in initiation fluctuations is a ladder or continuum of activity states. The timescale of initiation rate variation through this spectrum of activity states can be determined using an autocorrelation analysis of non-zero initiation rates. *Figure 3F* shows such a plot, which reveals the initiation rate fluctuates with an average timescale of around 5–6 min, although it is possible for a roughly constant initiation rate to be sustained for up to 15 min (*Figure 3—figure supplement 2*).

## Estimating fluctuations in elongation rate

To what extent can fluctuations in elongation rate contribute to the complexity of the ladder or continuum of transcriptional states? A transiently slower rate of elongation may cause a build-up in the number of polymerases on the gene and therefore contribute to non-Poisson variations in spot intensity. To test this possibility, we use simulations with three initiation rate states – with average initiation and elongation rates matched to the experimental estimates – to address whether the additional complexity shown experimentally beyond a three state model (*Figure 3D*) could be accounted for by adding elongation rate fluctuations to the system.

In addition to increasing the spot intensity, slower elongation rates lead to increased dwell time, which in FRAP measurements, would lead to the brighter spots taking longer to recover. However, the experimental data (*Figure 2D*) showed no observable difference in recovery between groups of different intensity. It remains possible that fluctuations in elongation rate may occur over a timescale which would not be resolved by FRAP. To investigate the timescale of dwell time fluctuations that might be invisible to FRAP, we developed simulations of the FRAP protocol, matching the experimental procedure as closely as possible. We incorporated temporal fluctuations in the polymerase elongation rate acting either globally (affecting every polymerase on the gene in the same way) or in a polymerase-by-polymerase manner. Since very little is known about the type of elongation

fluctuations possible in vivo, we implemented a system which switches randomly between 10 and 30 nucleotides/s, and varied the timescale of fluctuation (how long the system remains in either state).

The FRAP curves produced by individual simulated cells are treated in the same way as experimental data - dividing the cells into three groups based on their recovery intensity and rejecting cells showing no recovery – to determine the largest fluctuation timescale which shows no difference between intensity groups. As shown in *Figure 4A*, elongation rates fluctuating independently for each polymerase cannot produce any differences in the recovery curves. For the density of polymerases on the gene for typical spot intensities, fluctuations of individual polymerases are suppressed by catching up and being blocked by a polymerase immediately downstream. This results in a bulk polymerase elongation rate close to the lower speed of 10 nucleotides/s. Thus independent fluctuations in polymerase elongation cannot account for the fluctuations in spot intensity in the continuum model.

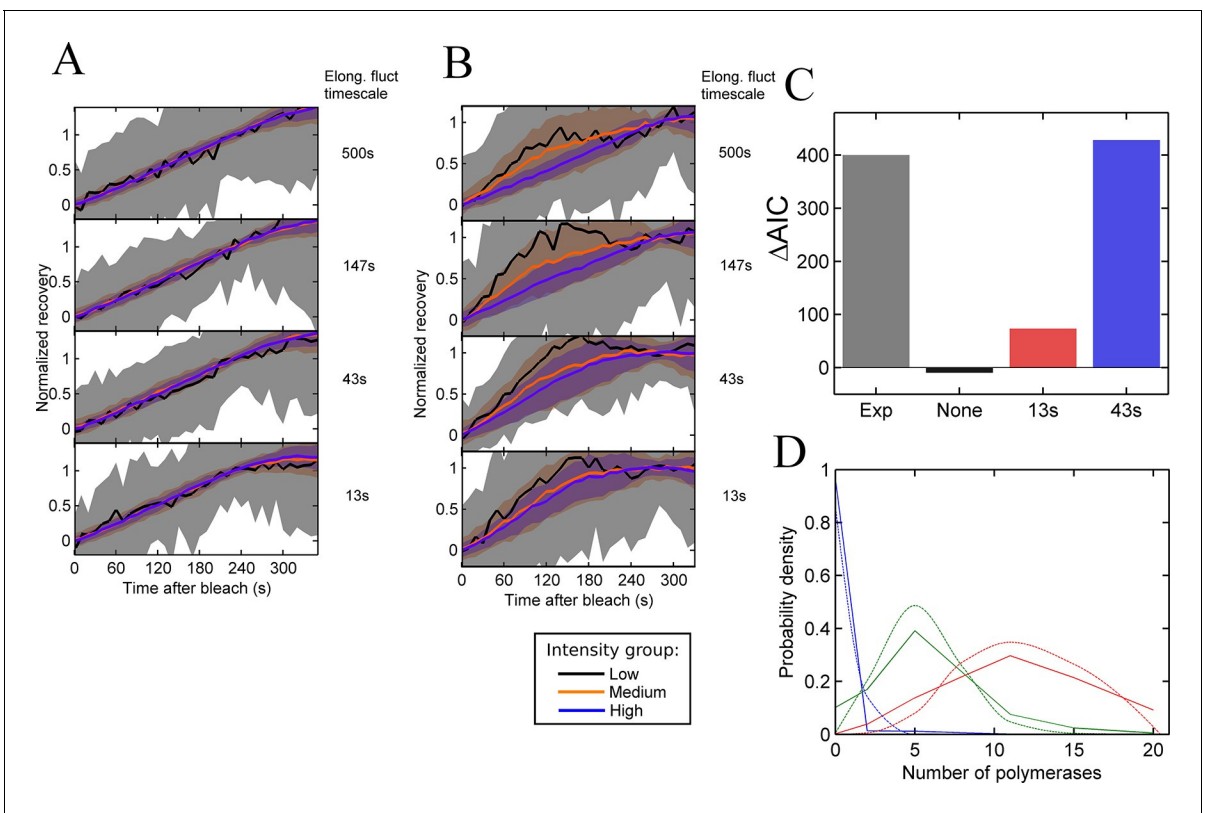

**Figure 4.** Testing the contribution of elongation rate switching to intensity fluctuations See also *Figure 4—figure supplement 1*. A and B Simulated FRAP measurements for a system with three states of initiation rate and two elongation rate states. Initiation rate dynamics are chosen to match those observed experimentally, while the timescale of elongation rate fluctuations is varied from 500 s (top panel) to 13 s (bottom panel) between 10 bases/s and 30 bases/s. In **A**, the elongation rate for each polymerase fluctuates independently from other polymerases, whereas in **B**, all polymerases move with a global fluctuating elongation rate. The simulated data are subdivided equally between three bins of low (black), medium (orange) and high (blue) spot intensity, as with the experimental data in *Figure 2D*. Differences between bins are only apparent with global fluctuations. Variability is shown with standard deviations. (**C**) Effects of elongation rate fluctuations on the 3-state simulation. The y-axis shows the increase in complexity produced by adding elongation fluctuations to a three-state simulation, compared with experimental results. Simulated data is slowly varying three-state initiations with fast-varying two-state elongations. Simulations with fast fluctuations (13 s) show a small improvement in fit above three states (red bar). Simulations with 43 s timescale elongation fluctuations (blue) show an improvement in fit comparable to experimental data (grey). (**D**) Polymerase distributions in three-state model fit for 3-state simulation with 43 s elongation fluctuations (solid, straight lines), compared with Poisson best fit (dotted, curved lines).

The following figure supplement is available for figure 4:

**Figure supplement 1.** Training a three-initiation-state model on simulated data with three initiation states and no elongation rate fluctuations (see main text).

In contrast, global fluctuations in the elongation rate give rise to different recovery curves for different spot intensities (*Figure 4B*), as the brightest spots tend to be caused by slow-moving polymerases staying at the transcription site for longer, while the intermediate spots have a bias towards faster moving polymerases. As in experiments, the lowest intensity recoveries show a high degree of noise, due to the small normalisation factor. As the timescale of the elongation fluctuations becomes shorter, the difference between the recoveries is reduced, as variability in the elongation rate is blurred out by faster transitions. The point at which the simulated data becomes consistent with the experimental traces (*Figure 2D*) is around the lower two panels of *Figure 4B*, that is, an elongation fluctuation timescale with an upper limit of around 10-40 s. What this effectively means is that gene-wide fluctuations in the elongation rate may be present on a timescale of up to around 40 s without being apparent in experimental FRAP recovery curves. The exact timescale depends also on the magnitude of the fluctuations, taken here to be around 50% of the mean elongation rate, nevertheless, some degree of elongation fluctuation may contribute to the temporal variation of the transcription spot intensity.

We then asked whether adding elongation fluctuations of the magnitude identified above to the three-state model is sufficient to produce the decrease in AIC value beyond 3 states seen in hidden Markov modelling. The difference, ΔAIC, between AIC (3-state) and the minimal AIC value (for a model up to 8-states) represents the additional complexity present in the data – essentially the information missing from a three-state fit. We applied our HMM fitting to simulations of the three-state model with different timescales of elongation fluctuation, and measured ΔAIC to estimate if the addition of elongation fluctuations is sufficient to recapitulate the additional complexity measured experimentally.

*Figure 4C* compares the ΔAIC value between a representative experiment and simulations of elongation fluctuations (adjusted to match experimental sample size). For no fluctuations, no improvement is seen as expected, since a three-state model successfully describes the simulated data. For 13 s elongation fluctuations a small increase in complexity is observed. For the 43 s timescale, the magnitude of the improvement is comparable to that observed experimentally. This suggests these elongation fluctuations can potentially recapitulate the variability in spot intensities observed in experiments, with two important caveats. Firstly, in the simulations the 43 s timescale data plateau at 5 states, rather than around 7–8 for the experimental data. Secondly, training a three-state model on experimental data yielded distributions of polymerase initiations in each state with variances greater than expected for Poisson distributions (*Figure 3E*). This super-Poissonian behaviour of the experimental data is not reproduced by training the same three state model on in silico elongation fluctuation data (*Figure 4D*, *Figure 4—figure supplement 1*), indicating further complexity in the experimental data not fully explained by the simple elongation fluctuations.

## Effects of perturbing the initiation rate

To test the effects of perturbing the initiation rate on transcriptional bursting, we generated point mutations in the TATA box of the *act5* promoter, T1A and A2C (*Figure 5A* and *Figure 5—figure supplement 1* and *2*), which have strong effects on transcriptional output in yeast genes and mammalian expression plasmids (*Patwardhan et al., 2009*; *Raser and O'Shea, 2004*). Both TATA mutant lines displayed a slight reduction in overall spot intensity (*Figure 5B*), with FRAP experiments suggesting a similar dwell time to wild type (*Figure 5—figure supplement 3*). Analysis of time-lapse experiments using the polymerase HMM framework found that the TATA mutants spend less time in medium and high initiation rate states, and more time at lower initiation rates, compared to control cells (*Figure 5C*). The overall timescale of initiation rate variability, measured through autocorrelation, was not substantially changed (*Figure 5D*) with both the wild-type and TATA mutations showing fluctuation timescales of several minutes. In addition, we observed no clear difference in switching rates between the inactive and active states. The rate of switching to the inactive state, *k (off)*, was unchanged between wild-type and TATA mutants (*Figure 5E*). The tendency of spots to appear, *k(on)*, showed a slight impairment in A2C mutants, although this was not statistically significant. This subtle effect might be also interpreted as the enhanced occupancy of active states of unobservably low intensity in the A2C mutation, rather than simply the absence of transcription. We then addressed the likelihood of the gene switching up or down in initiation rate based upon its current state (*Figure 5—figure supplement 4*). For all cell lines, both wild-type and TATA mutant, high initiation rates had a tendency to revert to lower initiation rates, and lower initiation rates had a

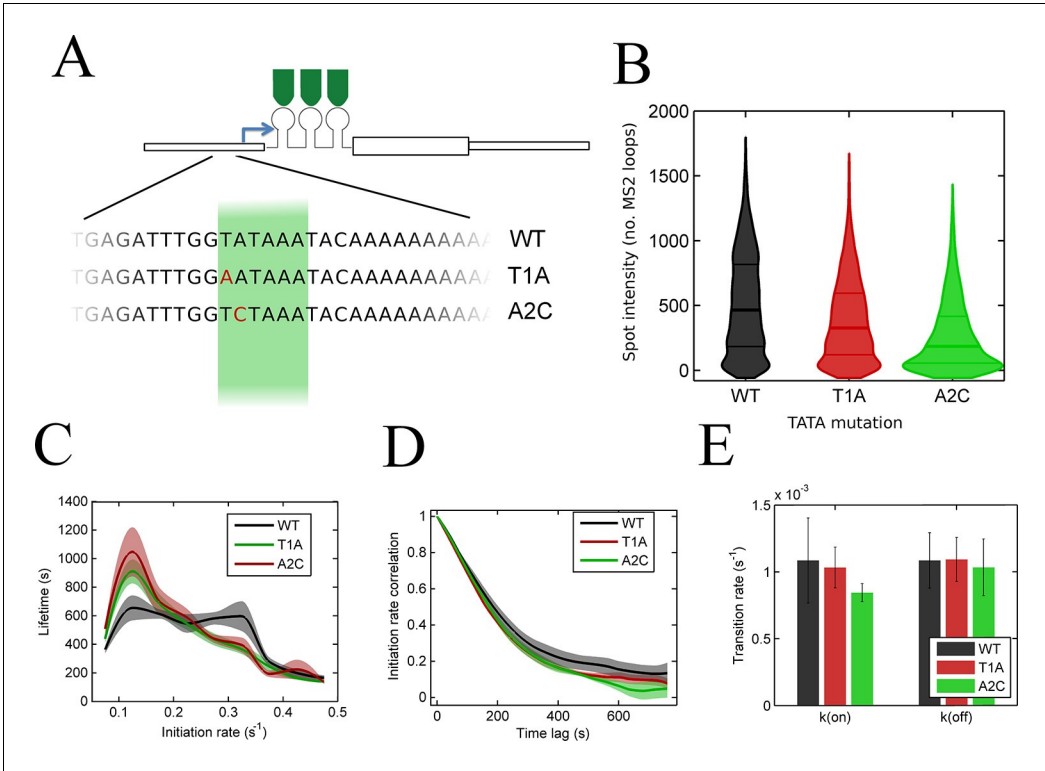

**Figure 5.** The TATA box influences access to the high activity states. See also *Figure 5—figure supplements 1–4*. (**A**) TATA box mutations studied for the *act5* gene. (**B**) Probability density function of transcription site intensity for TATA mutations T1A and A2C compared to WT. One of four biological replicates is shown. The reduction in intensity in the TATA mutations is slight, but significant (KS test: $p=10^{-58}$ for wt vs. T1A and $p=10^{-158}$ for wt vs. A2C). (**C**) Lifetime of constant initiation rate pulses in the active state, as a function of initiation rate for TATA mutants compared to control. The TATA mutants spend longer in lower initiation states and shorter durations at high initiation rates. The curves display mean and S.E.M. from 4 independent experiments (with 1686–6350 individual frames from 44–145 individual cell tracks, from each cell line, from each of the 4 replicates). We used grouped ratio t-tests to compare distributions, pooling the data based upon initiation rate. For low initiation rates ($<0.2\ s^{-1}$) gave $p=0.0083$ and $0.0015$ for T1A and A2C respectively. For high rates ($>0.25\ s^{-1}$) gave $p=3.5 \times 10^{-5}$ and $0.0011$. A breakdown of the data is contained in the Supplementary Material. (**D**) Timescale of initiation rate persistence, as measured by the decay of the autocorrelation of instantaneous initiation rate, is similar for TATA mutants and WT. (**E**) Estimated rates of transition from closed to open state (k(on)) and from open to closed state (k(off)). Values are average of 4 experiments. Error bars are S.E.M. Differences are all non-significant (p all >0.45).

The following figure supplements are available for figure 5:

**Figure supplement 1.** Example WT spot intensity traces.

**Figure supplement 2.** Example spot intensity traces for the A2C TATA box mutation cell line.

**Figure supplement 3.** Fluorescence recovery after photobleaching (FRAP) curves show no evidence for different RNA dwell times in the TATA mutants (T1A, A2C) compared to wild type (WT).

**Figure supplement 4.** Left - probability of increasing (dotted lines) or decreasing initiation rate (solid lines) as a function of initiation rate for the *act5*-MS2 wild type (WT) and TATA mutant cell lines.

tendency to revert to higher initiation rates. In the TATA mutants, the initiation rate is less likely to switch to a higher activity state, resulting in reduced time spent in high activity states. Together these observations suggest that perturbing the TATA box does not affect the duration or frequency of active states, but rather modulates the initiation rates that are possible.

## A continuum model of transcription

The standard analysis of transcriptional bursts using smFISH measurements can extract estimates of the burst size and burst frequency. Implicit in the assumptions of the random telegraph (two state) model is the idea that cells undergoing a burst in a particular gene all have the same underlying initiation rate. However, it seems unlikely, based on our data, that the active initiation rate will be constant over time during a burst, or that it will be the same for all cells in a population. Rather, the rate of initiation will depend on the binding of molecular factors, which turn over on the timescale of seconds to tens of seconds, based upon measured residence times of transcription factors (*Chen et al., 2014*; *Izeddin et al., 2014*). The cytoplasmic lifetime of mRNA means that these fast fluctuations are blurred out in static measurements, such as smFISH. In most such cases only broad differences in overall transcript content between cells can be determined, which accounts for the success of models with a few numbers of discrete states in fitting the data. Measurements using the MS2 system are integrated over the dwell time of the nascent RNA, a timescale of 2–3 min. The dynamic measurements can begin to resolve systematic variability in the initiation rate over time, which is difficult to assign to a small number of discrete states. Our proposed continuum model is summarized in *Figure 6A*. This model may also be approximated by a discrete state model with more states than can be effectively detected (*Figure 6—figure supplement 1*).

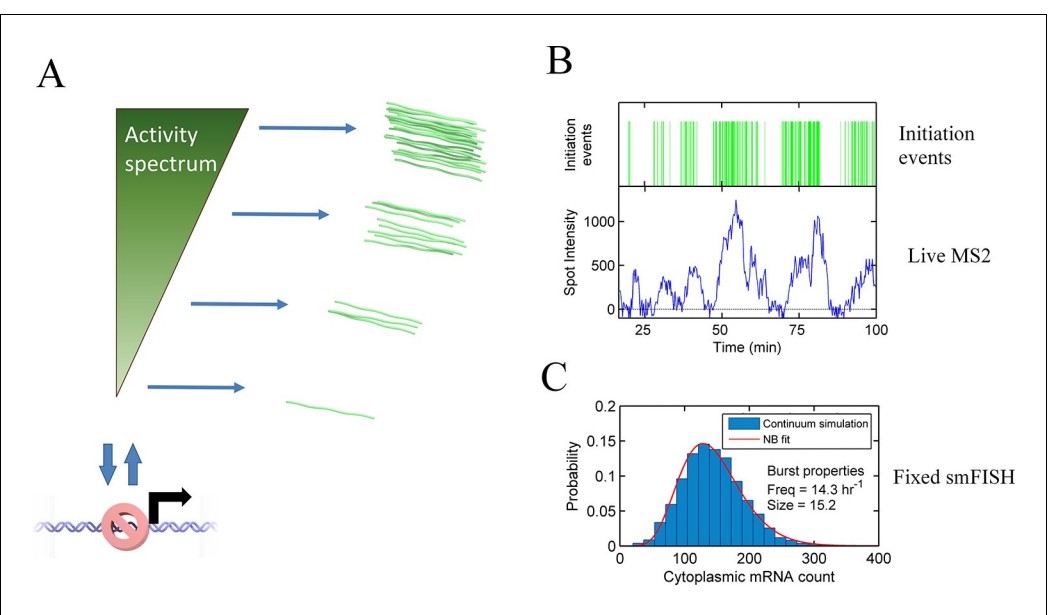

**Figure 6.** Continuum model. See also *Figure 6—figure supplements 1* and *2*. (**A**) Proposed continuum model. In addition to switches to and from a closed state on the timescale of around ten minutes, the initiation rate in the active state fluctuates on a shorter timescale. (**B**) Simulation of the continuum model, resulting in temporal variation in the initiation rate (upper, green spikes). The short integration time of MS2 measurements (the time for which RNA is retained at the transcription site) means fluctuations in the active state of the gene can be visualized (lower). (**C**) In simulated smFISH data (right), using the RNA production events from the continuum model (**B**) and a cytoplasmic RNA decay time of 40 min, the distribution is well described by a standard two state bursting (negative binomial, NB) model. The long lifetime of cytoplasmic RNA averages out the temporal fluctuations in the initiation rate.

The following figure supplements are available for figure 6:

**Figure supplement 1.** Potential mechanisms by which a continuum of activity (ii) may arise: (i) a ladder containing a large number of discrete states, each with a distinct initiation rate, caused by specific binding of transcription factors or epigenetic marks.

**Figure supplement 2.** Cartoon illustrating the continuum model and the predicted changes caused by TATA sequence modification.

Our continuum model may be consistent with a recent theoretical model (*Rieckh and Tkačik, 2014*) which proposed rapid switching between states as an improvement over the standard 2-state model in terms of information transmission (*Figure 6—figure supplement 1iii*). In this view the initiation rate fluctuations can be thought of as the fraction of time for which the factors required for initiation are bound to the promoter. The rate of binding depends on the spatiotemporal variability in local concentrations, producing a continuum of activities when integrated over the MS2 RNA dwell time. When the TATA box is mutated, the overall decreased initiation rate in the active state may reflect the reduced rate of binding of necessary transcription factors, and therefore reduced occupancy of the initiating state (*Figure 6—figure supplement 2*). Illustrating the potential equivalence of this rapid switching model with our continuum model, our data show that the optimal discrete 3-state model consigns polymerase initiations to non-Poissonian states (*Figure 3E*), which implies either multiple initiation rates within a state (in other words more than 3 states) or switching between the discrete states faster than the imaging frame interval.

Is this continuum view in conflict with the existing smFISH data on 2-state models? To answer this question, we simulated steady state levels of RNA derived from a gene where the binding rate varies continuously over time, overlaid with longer periods of inactivity (*Figure 6B*, upper). We incorporated autocorrelation-derived timescales and initiation rates extracted from the hidden Markov modelling together with a mRNA lifetime of 40 min, similar to measured lifetime values of *act5*-MS2 using actinomycinD-treated cells. The distribution of cytoplasmic mRNA molecules was well fit to a standard two-state bursting model (*Singer et al., 2014*), using a negative binomial function (see appendix) capturing primarily the long timescale activity intervals (*Figure 6C*). The shorter timescale of the MS2 measurements (*Figure 6B*, lower) is able to discriminate fluctuations within active regions. In other words, when measured in the steady state, using methods such as RNA FISH, a continuum of transcriptional states will appear to consist of far fewer states than can be detected using live cell approaches, highlighting the different timescales measured by different methods. The two-state model approximates the magnitude and broad timescale of transcriptional variability, although it should be noted that the bursting parameters extracted may not directly correspond to the physical quantities of burst size and frequency.

## Discussion

We have carried out in-depth measurements of the transcription dynamics of an endogenous, highly-expressed gene in *Dictyostelium*. This analysis has provided an insight into the dynamics of transcriptional regulation at short timescales, allowing models of the underlying mechanics to be discriminated. We interpreted fluctuations in the intensity of the transcription site in terms of the time-resolved rate of polymerase initiation, and found that the behaviour is not consistent with either constitutive, Poissonian activity or switching between a small number of discrete activity states. Instead, we observed a spectrum of activity states, characterised by an initiation rate varying over timescales of several minutes.

This spectrum of activity may be produced by very short periods of transcription initiation, where the activity lifetime or rate of reactivation is dependent on the time-varying local activity of molecular factors, or the gene locus switching between different topological conformations. When integrated over the retention time of nascent RNA at the transcription site, this 'microbursting' behaviour would give rise to a continuum of initiation rates. An alternative model might involve each cell having a distinct initiation rate in the active state, thereby producing a spectrum of initiation rates but with a simple two-state model for each cell (*Sherman et al., 2015*). This model is equivalent to a limiting case of our continuum model with infinitesimally slow variation in the activation rate, however, for *act5* we observed timescales of variation of minutes by autocorrelation or Gibbs sampling of initiation rates. The differences between our continuum view and models with each cell having it's own stable transcription rate may reflect differences in experimental system. Alternatively, the differences may arise because live cell analysis of nascent RNA dynamics allows the timescales of fluctuation to be directly extracted from the data, which is not possible from measurements of protein abundance or steady state transcript counts.

Bursts of transcriptional activity have been described in a wide range of systems (*Raj and van Oudenaarden, 2008*; *Chubb and Liverpool, 2010*). To investigate the molecular origins of the bursting phenomenon, we constructed a simple theory of Poissonian transcription of the MS2

system. Unexpectedly, this simple system with constant promoter activity predicts exponential distributions of transcriptional pulses and intervals, with the pulse duration determined not only by the promoter initiation rate but also by the frame interval of observations and threshold of spot detection. We confirmed the predictions of the theory using Monte Carlo simulations of the MS2-tagged gene. In general, these simulations provide a framework for studying fluctuations in transcription site intensity, incorporating stochastic events for polymerase initiation, transition to elongation, elongation and termination/release. Additional complexity could be added in the form of the probability of promoter-proximal pausing, premature termination or interactions between adjacent polymerases. This approach allows us to predict the change in the spot intensity due to adding or changing a system parameter, and therefore determine whether competing models can be distinguished experimentally. This is illustrated by our analysis of the fluctuations in the elongation rate, where, in comparison with experimental data, we estimated an upper limit for the timescales of fluctuations at around 40 s. Analysis of elongation rate changes suggested such fluctuations could not account entirely for the complexity of the spectrum of transcriptional states, although they could potentially contribute.

Interspersed with the active regions are periods of relative inactivity, where no transcription spot is visible. The number of steps involved in the reactivation of an inactive gene has been inferred from the distribution of the off-durations (*Suter et al., 2011*; *Molina et al., 2013*). Due to the current detection threshold of our system, a low level of basal transcription cannot be ruled out in this inactive state. Consistent with early views of bursting (*Raj and van Oudenaarden, 2008*), the longer periods of inactivity may reflect slower dynamics of the remodelling of chromatin, rather than binding of specific transcriptional regulators, based upon our perturbation of the core promoter. Mutation of the *act5* TATA box reduced the overall amount of transcription, primarily by reducing the amount of time spent at high initiation rates rather than changing the switching dynamics between the off and active states. Importantly, neither TATA box mutation we studied abrogated transcription entirely, as might be expected if the pre-initiation complex can no longer bind the promoter. A reduction in the duration of periods of high activity might suggest an impaired duration or frequency of binding, or disruption of normal promoter conformational switches in response to binding (*Gietl et al., 2014*). The strong effects of TATA mutations in yeast and plasmid systems (*Patwardhan et al., 2009*; *Raser and O'Shea, 2004*) may reflect more simple promoter architectures. In more complex systems, the potential for many inputs to transcriptional regulation will buffer the disruption of any single input (*Perry et al., 2010*).

It must be stressed that our quantitative analysis has been carried out on one gene, in steady-state conditions. The gene is strongly expressed and actin is usually put under the umbrella of 'housekeeping', which is a slight simplification as most of the *Dictyostelium* actin gene family, including *act5*, show some developmental regulation, at least at the transcript level (*Muramoto et al., 2012*; *Joseph et al., 2008*). A more strictly induced developmental gene might be expected to show more strict two state ON/OFF behaviour. However, even in cases with more prominent bursting we would argue that the initiation rate in the active state is likely to fluctuate over time and differ from cell to cell. This is supported by recent studies on mammalian transcriptional induction by serum and growth factors, which also suggest more complexity than a standard 2-state model provides, with evidence from luciferase reporter fluctuations (*Molina et al., 2013*) and measurements of nascent RNA by smFISH indicating modulation of the transcription rate within the ON state (*Senecal et al., 2014*). A more recent smFISH study quantifying nascent RNA revealed modulation of burst size and frequency by cell size and cycle stage, respectively (*Padovan-Merhar et al., 2015*). Whilst we have argued here for a more complex view of transcriptional regulation, it must be considered that the *act5* promoter is less than 700 bp long, the gene contains no introns and around 60% of the *Dictyostelium* genome encodes protein (*Eichinger et al., 2005*), and so provides little scope for long range regulatory interactions. In the light of these features, we suggest the spectrum of states is likely to have considerably more scope for complexity in a mammalian cell. The continuum of states we infer is likely to be a more realistic view of gene activity fluctuations than the standard views of a small number of fixed discrete states. There are perhaps a hundred different proteins involved in a standard eukaryotic transcription reaction, even ignoring the components of the chromatin template. The likely configurational complexity, in addition to the potential for modulation by protein modification and nuclear context, seems consistent with the continuum view.

## Materials and methods

### Molecular biology and cell line generation

For targeting of MS2 repeats into the actin 5 gene, we utilized a BsrGI-SpeI restriction fragment containing 24 MS2 repeats (1.3 kb) upstream of the blasticidin resistance (bsr) cassette (*Faix et al., 2004*). The resistance cassette is flanked by loxP sites for CRE-mediated removal of the marker, allowing transcription to terminate at the natural 3' sequence of the gene. For the 5' tagging of *act5*, the MS2-bsr was cloned between a promoter fragment of the gene (-680 to +21) and a gene fragment (+108 to +1313), using BsrGI and SpeI sites. Cells derived using this 5' tagging vector were used for all experiments described in this paper, unless indicated otherwise. For the gene replacement vector, the MS2-bsr was cloned between the same promoter fragment and a fragment from the 3' coding sequence and terminator of the gene (+1092 to +1899). For the 3' targeting vector, we used this same 3' region combined with a 5' region derived from the act5 coding sequence (+259 to +1113). The ATG corresponds to +1. The translational STOP, TAA is at +1129. The first clear polyadenylation motif (AATAAA) starts at +1193. Sequences were checked at each cloning step to ensure plasmid stocks retained the correct sequences (as specified by dictyBase). For TATA mutations, the T1A (AATAAAT) and A2C (TCTAAAT) mutant promoter fragments were generated by gene synthesis, then spliced into the wild-type promoter sequence using the BstEII site upstream of the TATA box, prior to inserting the MS2-bsr fusion, again using BsrGI and SpeI sites at the same positions.

Targeting fragments were released from cloning vectors by digestion with polylinker enzymes ClaI and NotI. These targeting fragments were transformed into a *Dictyostelium* AX3 clone previously engineered to express a red fluorescent nuclear marker, H2Bv3-Cherry, under the control of the endogenous promoter of the *rps30* gene. Targeted clones were identified by PCR, then genomic DNA Southern blotted to ensure MS2 repeat integrity and single correct insertions in the targeted clones. Correct targeting of TATA mutations to the *act5* promoter was checked by sequencing of PCR products from recombinant clones. Positive clones were then transiently transfected with a plasmid expressing the CRE recombinase, to remove the bsr cassette, allowing the MS2 RNA to fall under the control of the natural *act5* terminator. Clones were then transformed with an extrachromosomal vector, based on pDM1096 (from Dr. Douwe Veltman) expressing the MCP-GFP fusion protein, to permit detection of nascent RNA in living cells. Selection used 20 μg/ml G418.

### Live imaging of MS2 cell lines

Cell culture preparation for imaging was carried using cells grown in HL5 medium (FORMEDIUM) supplemented with penicillin+streptomycin. 20 μg/ml G418 selection was added 72 hr after thawing frozen stocks. 18 hr prior to imaging, cells were split into imaging chambers (NUNC LabTek-II) at the appropriate density for imaging the next day at around 20% confluency, drug selection was removed and HL5 was replaced with imaging medium (75% LoFlo medium (Formedium), 10% FBS, 15% HL5). Imaging media was refreshed 1.5 hr before imaging. Imaging was performed using an UltraView Vox spinning disc confocal microscope. Objective, laser lines, camera (Hamamatsu C9100-13 EM-CCD) settings, laser powers and exposure times were optimized to minimize photobleaching and ensure negligible photo-toxicity (measured in terms of the average transcription spot intensity - in trials we found transcription spots were attenuated before reduced cell motility or cell rounding is observed). Data were analysed using custom-built software integrating both cell tracking and spot detection. Code can be accessed at http://www.ucl.ac.uk/lmcb/sites/default/files/Corrigan2016MatlabFiles.zip.

### Single molecule RNA FISH calibration of live intensities

Cells were imaged live as described above for a single time point and then immediately fixed and prepared for smFISH measurements following the procedure outlined in *Raj et al. (2006)*, using a single probe (CATGGGTGATCCTCATGT; Biosearch, Petaluma, CA) against the repeated spacer between the each MS2 loop sequence, end-labelled with Quasar 570 fluorophore. Nuclei were co-stained with DAPI. Image stacks were acquired using the spinning disc using exposure times of 3 s for single molecule sensitivity. Individual cytoplasmic RNA molecules and nascent transcription sites were detected using FISH-quant software v2 (*Mueller et al., 2013*), augmented with a custom-written dual-threshold algorithm to segment nuclei and cytoplasm of cells based on the DAPI signal.

## Simulation and modelling methods

A Monte Carlo simulation framework for the MS2 system was constructed using custom-written tools in MATLAB. For simulation of FRAP experiments, binding and potential unbinding of MCP was simulated with first order kinetics. Tools for analysis of $n^{th}$-order hidden Markov models, including the forward-backward and Baum-Welch algorithms and Gibbs sampling, were custom-written using MATLAB. Full details of the simulation and probabilistic analysis procedures are presented in the appendix.

## Additional information

### Funding

| Funder | Author |
|--------|--------|
| Wellcome Trust | Jonathan R Chubb |
| Biotechnology and Biological Sciences Research Council | Jonathan R Chubb |

The funders had no role in study design, data collection and interpretation, or the decision to submit the work for publication.

### Author contributions

AMC, Conception and design, Acquisition of data, Analysis and interpretation of data, Drafting or revising the article, Contributed unpublished essential data or reagents; ET, DC, Acquisition of data, Drafting or revising the article; JRC, Conception and design, Analysis and interpretation of data, Drafting or revising the article, Contributed unpublished essential data or reagents

### Author ORCIDs

Jonathan R Chubb, http://orcid.org/0000-0001-6898-9765

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

# Appendix 1 Supplementary experimental procedures

## 1.1 FRAP data collection and analysis

Fluorescence recovery after photobleaching (FRAP) experiments were performed following previously described procedures (*Muramoto et al., 2012*). Briefly, a cell with an active transcription site was identified and a single z-stack captured prior to bleaching. Bleaching parameters (bleach size, number of bleaching cycles, bleach power) were optimized empirically to give reliable bleaching, reproducible recovery and minimal photodamage. Two seconds after bleaching, z-stacks were acquired with a frame interval of 10 s for 5 min. The location of bleached spots is difficult to robustly measure automatically, therefore after automated segmentation and spot detection, spot locations were verified manually using a custom-built interface before spot intensity calculation. For each cell, the spot intensity and the relative local level of background fluorescence was recorded for every frame.

## 1.2 Estimation of elongation and termination times from FRAP data

To estimate the RNA dwell time, we use a robust weighted averaging scheme to generate a smooth recovery curve, excluding outliers in an unbiased manner. Traces were weighted according to the difference from the mean, averaged over all frames of each trace (one weight value per cell), and the weighted average recovery calculated.

Weights were recalculated using distance from the new average recovery, and this process was iterated until the weights no longer changed with a relative tolerance of $10^{-3}$. This approach assigns low weights to cells that behave significantly differently to the majority of cells, for example cells where the transcription spot disappears or does not recover. The number of such cells is in agreement with the observed frequency of transcription becoming inactive in a five minute window.

The average recovery curve is then fit to a model recovery which takes as inputs the time to produce a full complement of loops, $t_{elong}$, the time spent with a full complement of loops, $t_{full}$, and the normalised recovery of the background fluorescent MCP level, $f(t)$. Approximating the gene to be uniformly occupied by polymerases, the rate of change of transcription spot intensity is calculated as the balance between the gain of newly produced loops being bound by unbleached MCP ($A$) and the loss of unbleached loops when RNA leaves the site, from fluorophores binding newly produced loops ($B$). If the bleach does not completely remove the spot fluorescence, a further source of fluorophore loss arises from the pre-existing, partially bleached RNA leaving the transcription site $C$). This contribution is proportional to the unbleached spot fraction $\Gamma$. By calculating the number of loops per polymerase as a function of position on the gene, the rate of intensity loss due to incompletely bleached RNA leaving the TS is:

$$\frac{dC}{dt} = \Gamma \times \begin{cases} 1 & \text{for } t < t_{elong} \\ 1 - \dfrac{t - t_{full}}{t_{elong}} & \text{for } t_{elong} < t < (t_{elong} + t_{full}) \\ 0 & \text{otherwise} \end{cases}$$

The rate of intensity loss due to newly produced loops and unbleached fluorophore leaving the TS depends on the level of the background fluorescence at the point that each loop was produced (i.e. loops produced shortly after the bleaching event will be more likely to be bound by bleached MCP, since the background level takes a finite time to recover). Therefore when an RNA leaves at time t, we integrate the background fluorescence over the range of times when the loops of this RNA were produced:

$$\frac{dB}{dt} = \int_0^t f(t')p(t-t')dt',$$

where:

$$p(\tau) = \begin{cases} \dfrac{1}{t_{loops}} & \text{for } t_{term} < \tau < (t_{loops} + t_{term})) \\ 0 & \text{otherwise} \end{cases}$$

The rate of intensity increase due to newly produced loops being bound by unbleached fluorophore depends on the recovery of the background level of unbleached fluorophore, $f(t)$:

$$\frac{dA}{dt} = f(t)$$

The simplicity of this expression stems from the assumption that polymerases are uniformly spread over the gene. The time evolution of the transcription site intensity is then calculated by numerically integrating the rates of fluorescence loss and gain:

$$I(t) = \Gamma + \frac{1}{\frac{1}{2}t_{loops} + t_{full}} \int_0^t \left( \frac{dA}{dt'} - \frac{dB}{dt'} - \Gamma \frac{dC}{dt'} \right) dt'$$

Least-squares optimization is then performed to calculate the values of $t_{elong}$ and $t_{full}$ which are consistent with the observed recovery, using the observed background level to estimate the fraction of unbleached MCP available. In practice, the value of ($t_{elong} + t_{full}$) can be calculated robustly.

## 1.3 Details of automatic tracking algorithm

To track cells from one frame to the next, we use the coordinates of the nuclear centroid as the spatial coordinates as the nucleus tends to more robustly segmented than the cell outline. For dense fields of view a greedy algorithm, which assigns a cell to the closest cell in the previous frame, performs very poorly. Instead, we minimize the global sum of squared displacements from one frame to the next, adding penalties for unmatched cells in either frame. Because of the flux of cells entering and leaving the field of view, we reduce the unmatched cost as the coordinate approaches the edge of the field of view. This modification prevents a cell leaving the field of view being matched with a nearby cell entering in the next frame. The optimal configuration of frame-to-frame matches is found using the Munkres algorithm (**Munkres, 1957**). To facilitate tracking of highly motile cells, where the frame-to-frame displacement can be comparable to the cell-cell separation, where necessary we include the mCherry and GFP fluorescence levels within the nucleus as additional coordinate dimensions. While differences in x- and y-coordinates are weighted quadratically, we use an alternative functional form for intensity differences, reflecting the increased measurement error for the fluorescence compared with the centroids. The function used has no cost for small differences in intensity, and becomes saturated at large intensity differences. This saturation value prevents a noisy intensity measurement, caused for example by poor segmentation, adversely influencing the matching.

A final consideration is that there may be several matching configurations with very similar global costs, and the configuration with the lowest absolute cost might not be the ground truth, due to high cell motility. To find matches which are robust and do not change in slightly sub-optimal configurations, we add an amount of randomness to each element of the cost matrix and recalculate the matches. This is repeated a large number of times, and only

matches which appear in greater than a threshold fraction of configurations are recorded. Although this can lead to an increase in false negatives, trackable cells not being matched, it is an important step to ensure that false positives, incorrect matches giving rise to anomalous transcription dynamics, are minimized. Therefore, we set the confidence threshold to a large value such as 0.8 or 0.9; cell-frames which do not meet this level are flagged for efficient manual tracking using a custom designed user interface written in MATLAB (The Mathworks, Natick, MA).

## Appendix 2 in silico dynamic transcription framework

### 2.1 Stochastic simulation methods

The MS2 system was constructed *in silico*, beginning with probabilistic modelling of polymerase initiation, elongation and termination using a modification of the well-known Gillespie algorithm (**Gillespie, 1977**). In the first stage, the promoter is modelled as switching across a 'ladder' of discrete initiation rate states, with single, exponentially-distributed steps for switches in the initiation rate. Typically, transitions are allowed only between adjacent or next-adjacent states. Using the generated switches in initiation rate, polymerase dynamics are simulated using single rate limiting steps for polymerase binding and single nucleotide elongation steps. Optionally, additional steps for the transition from transcription initiation to elongation, and for termination, poly-adenylation and RNA cleavage, can be included. The time for a cleaved RNA to leave the transcription site is typically in the timescale of seconds, and can be subsumed into the termination time. Polymerases that move within a specified 'blocking' distance of a downstream polymerase cannot make an elongation step until the downstream polymerase (in the 3' direction) has moved. Similarly, until a polymerase has moved more than the blocking distance from the binding site, the rate of an initiation event is set to zero.

### 2.2 Fluorescence and bleaching

For comparison with experiments, the binding of fluorescent MCP to the nascent RNA loops and measurement of the transcription site intensity are modelled. If the loop binding is assumed to be fast and irreversible (**Maiuri et al., 2011**), then the measurement error can be approximated by addition of random Gaussian noise. For simulation of photobleaching experiments, the concentration and diffusivity of the fluorophores is included. When the bleaching event is triggered, each bound MCP is bleached with a probability depending on the laser strength. Bleaching of the background MCP level is modelled with a Gaussian depletion profile centred on the transcription site to allow the time evolution of the fraction of fluorescent MCP at the site to be calculated analytically as a Gaussian solution of the 3D diffusion equation. For this purpose, the nuclear boundary and the finite number of fluorophores in the cell are neglected.

To calculate the recovery of the transcription spot, the total concentration of MCP is assumed to be constant over time, with the fraction of unbleached MCP recovering over time as described above. For simplicity it is assumed that the concentration of MCP (both bleached and unbleached) is not affected by RNA binding and unbinding events. We calculate the probabilities of an unbound loop becoming bound by bleached and unbleached fluorophores, and of a bound loop experiencing an unbinding event during one timestep, using estimated values for binding and unbinding rate constants and the MCP concentration and assuming first order reaction kinetics.

### 2.3 Correcting for polymerase blocking distance

In our model of MS2 transcription, a polymerase must move away from the binding site by a threshold distance before a subsequent polymerase can attach. This effectively means that polymerase attachment is no longer a Poisson process. However, by considering the distribution of time interval between successive initiations, a correction factor can be applied to calculate the effective initiation rate.

In a Poisson process with mean rate $\lambda$ the time interval between successive events has an exponential distribution with mean $\tau = 1/\lambda$. To correct for the polymerase blocking distance,

$D_{block}$, we modify the interval distribution to incorporate the probability that the binding site has been cleared by time $t$. For a polymerase taking single exponential steps, each of time $1/v_{el}$, where $v_{el}$ is the polymerase elongation rate, the time taken to take $D_{block}$ steps is approximated by:

$$P(\text{clearance time} = t) = \mathrm{N}(\mu = D_{block}/v_{el}, \sigma^2 = D_{block}/v_{el}^2),$$

by combining random variables using a Gaussian approximation. This expression can be used to modify the interval distribution:

$$P\left(\text{interval} = t'\right) \alpha \exp\left(-v_{el}t'\right).P\left(t' > \text{clearance time}\right)$$

However to a very good approximation we can use a deterministic clearance time of $D_{block}/v_{el}$ to calculate the correction to the initiation rate:

$$P(\text{interval} = t') = \begin{cases} exp\left(-v_{el}(t + \dfrac{D_{block}}{v_{el}})\right) & \text{for } t' > D_{block}/v_{el} \\ 0 & \text{otherwise} \end{cases}$$

The average time interval between polymerase initiations is given by:

$$\langle\text{interval}\rangle = \int P(\text{interval} = t')dt' \tag{2.1}$$

$$= \frac{D_{block}}{v_{el}} + \frac{1}{\lambda} \tag{2.2}$$

Therefore the effective initiation rate is given by:

$$\lambda_{eff} = \frac{1}{\frac{D_{block}}{v_{el}} + \frac{1}{\lambda}} \tag{2.3}$$

$$\lambda_{eff} = \frac{\lambda}{1 + \frac{\lambda D_{block}}{v_{el}}} \tag{2.4}$$

## Appendix 3 Stochastic theory of transcription pulses

## 3.1 Poisson distribution of polymerase initiations

### 3.1.1 Autocorrelation of pulse intensity

The shape of the fluorescence intensity measured over time for a single polymerase (assuming that fluorophore number scales directly with number of stem loops) can be approximated by a linearly increasing region where the stem loops are being transcribed, followed by a plateau when the polymerase traverses the remainder of the gene and waits for the transcript to be cleaved, before a sharp drop to zero when the RNA and attached fluorophores diffuse away from the transcription site. The approach of Larson *et al* (*Larson et al., 2011*) calculates the arrival time of each stem loop, and in doing so allows for stochasticity in the stepping time of the polymerase over each stem loop. Our simple model is applicable in the case where polymerases are sufficiently processive that the rate of loop production is approximately deterministic:

$$
n_{loops} = f(t) = \begin{cases} \dfrac{NT}{x}, & 0 < t < x; \\ N, & x < t < y \\ 0 & \text{otherwise} \end{cases}
$$

The unnormalized autocorrelation for $\tau < x$ is given by:

$$
G(\tau) \;=\; \int f(t)f(t+\tau)dt \tag{3.1}
$$

$$
= \int_0^{x-\tau} \left(\frac{Nt}{x}\right)\left(\frac{N(t+\tau)}{x}\right) dt + \int_{x-\tau}^{x} \frac{Nt}{x} N dt + \int_x^{y-\tau} N^2 dt \tag{3.2}
$$

$$
= \frac{N^2}{x^2}\left[\frac{t^3}{3}+\frac{t^2\,\tau}{2}\right]_0^{x-\tau} + \frac{N^2}{x^2}\left[\frac{t^2}{2}\right]_{x-\tau}^{x} + N^2(y-\tau-x) \tag{3.3}
$$

$$
= \frac{N^2}{x^2}\left(\frac{(x-\tau)^3}{3}+\frac{\tau(x-\tau)^2}{2}\right) + \frac{N^2}{2x}\left(x^2-(x-\tau)^2\right) + N^2(y-\tau-x) \tag{3.4}
$$

$$
= N^2\left(y-\frac{1}{2}\tau-\frac{2}{3}x-\frac{\tau^2}{2x}+\frac{\tau^3}{6x^2}\right) \tag{3.5}
$$

For $x < \tau < (y - x)$:

$$
G(\tau) \;=\; \int_0^x \frac{Nt}{x} N dt + \int_x^{y-\tau} N^2 dt \tag{3.6}
$$

$$
= \left[\frac{N^2 t^2}{2x}\right]_0^x + N^2(y-\tau-x) \tag{3.7}
$$

$$= N^2 \left( y - \frac{x}{2} - \tau \right) \tag{3.8}$$

For $(y - x) < \tau < y$:

$$G(\tau) = \int_0^{y-\tau} \frac{N^2 t}{x} dt \tag{3.9}$$

$$= \left[ \frac{N^2 t^2}{2x} \right]_0^{y-\tau} \tag{3.10}$$

$$= \frac{N^2 (y-\tau)^2}{2x} \tag{3.11}$$

The autocorrelation function can be normalized in a number of ways depending on the physical interpretation required. Typically the chosen denominator is the variance of the signal, which is calculated as:

$$\mathrm{Var}[n_{loops}] = E[n_{loops}^2] - (E[n_{loops}])^2 \tag{3.12}$$

$$\langle f^2 \rangle = \frac{1}{y} \left[ \int_0^x \left( \frac{Nt}{x} \right)^2 dt \int_x^y N^2 dt \right] \tag{3.13}$$

$$= \frac{1}{y} \left( \frac{N^2}{x^2} \cdot \frac{x^3}{3} + N^2 (y-x) \right) \tag{3.14}$$

$$= N^2 \left( 1 - \frac{2x}{3y} \right) \tag{3.15}$$

$$\langle f \rangle = \frac{1}{y} \left[ \int_0^x \frac{Nt}{x} dt \int_x^y N dt \right] \tag{3.16}$$

$$= \frac{1}{y} \left( \frac{N}{x} \cdot \frac{x^2}{2} + N(y-x) \right) \tag{3.17}$$

$$= N \left( 1 - \frac{x}{2y} \right) \tag{3.18}$$

$$\mathrm{Var}[n_{loops}] = N^2\left(1-\frac{2x}{3y}\right) - N^2\left(1-\frac{x}{2y}\right)^2 \tag{3.19}$$

$$= N^2\left(1-\frac{2x}{3y}-1+2\cdot\frac{x}{2y}-\frac{x^2}{4y^2}\right) \tag{3.20}$$

$$= N^2\left(\frac{x}{3y}-\frac{x^2}{4y^2}\right) \tag{3.21}$$

$$= \frac{N^2 x}{12y}\left(4-\frac{3x}{y}\right) \tag{3.22}$$

When normalized by the variance, the autocorrelation lies in the range $-1$ to $+1$, and is equivalent to the correlation coefficient between the spot intensity at a given time and the intensity a time $\tau$ later. We will use this calculation below in estimating the bivariate intensity distribution $P(I(t) = x, I(t+\tau) = y)$. Alternatively, the square of the mean can be used as normalization. This is frequently used in fluctuation analysis to study the relative size of deviations and to estimate the effective number of independent contributions to a signal. Larson *et al.* utilized this approach to estimate the number of polymerases contributing to the fluctuating spot intensity.

### 3.1.2 Stem loop distribution

The calculation of spot frequency is complicated by the fact that the contribution (measured in number of stem loops) of a polymerase varies from zero at the start of the gene to 24 at the end. Thus for a given threshold, the number of polymerases required to give a detectible spot is not fixed. Instead, the number of stem loops must be calculated explicitly.

The number of polymerases which have, for example, 3 stem loops transcribed is given by the number of polymerases which initiated within a short window of time. More recently initiated polymerases will have transcribed fewer stem loops, while those initiated a longer time ago will have more.

$$\text{Number of loops} = 1\times Pois\left(\frac{\lambda L}{Nv}\right) + 2\times Pois\left(\frac{\lambda L}{Nv}\right) + 3\times Pois\left(\frac{\lambda L}{Nv}\right) + \cdots + N\times Pois\left(\frac{\lambda L}{Nv}\right)$$

The Poisson distributions can be combined using a Gaussian approximation to calculate the mean and variance:

$$\langle n_{loops}\rangle = (1+2+\cdots+N)(\frac{\lambda L}{Nv}) \tag{3.23}$$

$$= \frac{1}{2}N(N+1)\frac{\lambda L}{Nv} \tag{3.24}$$

$$= \frac{(N+1)\lambda L}{2v} \tag{3.25}$$

$$\mathrm{Var}[n_{loops}] = (1 + 4 + 9 + \cdots + N^2)\left(\frac{\lambda L}{Nv}\right) \tag{3.26}$$

$$= \frac{N(N+1)(2N+1)}{6}\left(\frac{\lambda L}{Nv}\right) \tag{3.27}$$

$$= \frac{(N+1)(2N+1)\lambda L}{6v} \tag{3.28}$$

The coefficient of variation (CV) is defined as the ratio of the standard deviation and the mean. Because the value is dimensionless, the CV for the number of stem loops is expected to be equal to that of the GFP intensity. Therefore, the CV value does not depend on the scaling factor of GFP intensity units per stem loop.

$$CV = \frac{(\mathrm{Var}[n_{loops}])^{1/2}}{\langle n_{loops}\rangle} \tag{3.29}$$

$$= \frac{\sqrt{\frac{(N+1)(2N+1)\lambda L}{6v}}}{\frac{(N+1)\lambda L}{2v}} \tag{3.30}$$

For the case where there is a length of gene after the last stem loop, where the termination time is non-negligible or generally where the loops are not uniformly spaced, the equations above are not valid. However, the mean and variance of the Gaussian approximation can still be calculated numerically using the estimated relative time at which each loop is produced after polymerase attachment. In this case the expected number of polymerases with $k$ loops is a Poisson variable depending on the length of the window in time for which a polymerase has $k$ loops, defined as $\Delta t_k = t_{k+1} - t_k$:

$$\langle n_{loops}\rangle = \sum_{k=1}^{N} k \times [\Delta t_k \lambda] \tag{3.31}$$

$$\mathrm{Var}[n_{loops}] = \sum_{k=1}^{N} k^2 \times [\Delta t_k \lambda] \tag{3.32}$$

From the distribution of stem loop number at the transcription site, further properties such as the spot frequency and mean pulse intensity can be calculated for a given detection threshold.

## Spot frequency

The spot frequency is the probability that the number of stem loops at the transcription site is greater than the threshold:

$$f_{spot} = P\left(N(\mu, \sigma^2) > \phi_{loops}\right) \tag{3.33}$$

$$\mathrm{Let} \quad z = \frac{\phi_{loops} - \mu}{\sigma} \tag{3.34}$$

$$= \frac{1}{2}\left(1 + \mathrm{erf}\left(\frac{-z}{\sqrt{2}}\right)\right) \tag{3.35}$$

### 3.1.3 Distribution of pulse durations

#### Naive model

To maintain a pulse of a given threshold, on average a given minimum number of polymerases must be attached to the gene. Thus in order for a pulse to continue, there must be a new initiation within the average time taken for the final polymerase to be cleaved.

$$P(\text{Pulse ends}) = P\left(\text{no initiations in time } \frac{L}{vn_p}\right) \tag{3.36}$$

$$= P\left(Pois\left(\frac{\lambda L}{vn_p}\right) = 0\right) \tag{3.37}$$

$$= \exp\left(-\frac{\lambda L}{vn_p}\right) \tag{3.38}$$

Thus for timescales greater than $\frac{L}{vn_p}$, the pulse length has a geometric distribution, with mean duration:

$$\langle \tau \rangle = (timestep) \times \frac{1}{P(\text{Pulse ends at next timestep})} \tag{3.40}$$

$$= \frac{L}{vn_p}\exp\left(\frac{\lambda L}{vn_p}\right) \tag{3.41}$$

This value is an estimate only, since deviations from the steady state will persist for the dwell time of a polymerase.

#### Bivariate gaussian approximation

For a Poisson promoter, the spot intensity fluctuates over time but the distribution of spot intensities can be characterised by a static distribution, approximated by a Gaussian with properties defined by **equations 3.31** and **3.32**. To understand the change in spot intensity from one frame to the next, we considered the joint intensity distribution $P(I(t), I(t + t_{frame}))$ (JID) as a correlated bivariate Gaussian distribution. The correlation is calculated from the autocorrelation theory in section 3.1.1 and depends on the time at which each loop is produced and the frame interval. From this simple approximation, one predicts that pulse durations will be exponentially distributed, since the JID is constant over time and does not depend on the history of the intensity, that is, for how long the spot has been present. For any exponential distribution (or geometric distribution for discrete frames), the characteristic timescale, $\lambda$ is determined by the probability of stopping.

$$\lambda = \frac{1}{P(\text{off})}$$

For transcription spots, the stopping probability means the probability of no detectible spot in a frame, given that there was a detectible spot in the previous frame, that is $P(I(t + t_{frame}) < I_{thr}|I(t) > I_{thr})$. For a correlated two-dimensional Gaussian this probability cannot be calculated analytically, therefore we calculate the value numerically by sampling the JID over a fine grid to calculate the probability as:

$$P(off) \ = (I(t + t_{frame}) < I_{thr} | I(t) > I_{thr}) \tag{3.42}$$

Therefore, our pulsing theory predicts exponentially distributed pulses (and equivalently, intervals) for a constantly active Poisson promoter, where the average duration depends on the initiation rate, the time between polymerase intiation and production of each stem loop, and the frame interval. Therefore an exponential pulse distribution does not imply bursty transcription but simply suggests that the probability of a spot disappearing between one frame and the next is constant and does not depend on how long the spot has been present. We found that for Poissonian transcription the exponential pulse length is produced by a balance between polymerase initiation and retention; in simple terms, a pulse ends if MCP-rich polymerases reaching the end of the gene and being cleaved are not replaced by newly transcribed MCP-bound stem loops. This exponential distribution is a feature of stochastic transcription initiation and therefore does not necessarily reflect switches in gene activity, with the pulse length depending strongly on the imaging protocol.

## 3.2 Two state model of transcription

In the case of a two-state model of promoter activity, in addition to Poisson initiations in the active state, the promoter switches stochastically to and from an inactive state with no initiation. Thus a spot can disappear from stochastic fluctuations or gene switching, and hence P(off) and the pulse duration need to be modified accordingly. To calculate the joint intensity distribution for a two-state model, we break the gene into regions, corresponding to the distance moved by a polymerase in one frame, and build the JID combinatorically from components where the gene was off or on for each segment (*Appendix 3—figure 1*).

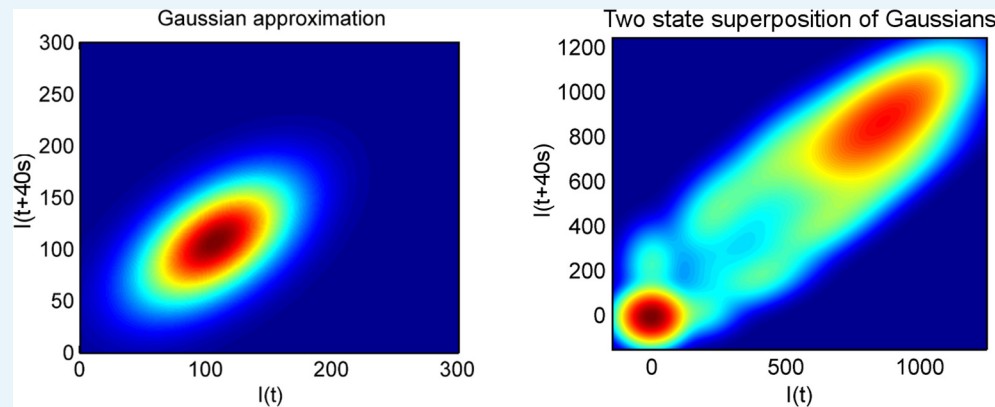

**Appendix 3—figure 1.** Using a joint intensity distribution to calculate pulse duration. The heatmaps show the probability of the spot intensity in one frame (x-axis) and the spot intensity in the next frame (y-axis), for a one-state (Poisson) model of constant initiation rate (left), and a two-state model (right), which switches randomly to and from an inactive state.

## Appendix 4 Hidden markov modelling of polymerase dynamics

### 4.1 Introduction to hidden markov models and methods

Hidden Markov models represent a discrete Bayesian approach to inferring the dynamics of an underlying (or hidden) state, which cannot be directly observed, from a sequence of observable emissions. In this respect, they are well-suited for studying transcription, where the transcription spot intensity fluctuates over time and is related probabilistically to the underlying configuration of nascent polymerases. The uses and methods of hidden Markov models have been described in detail previously by a large number of texts, and so we will focus on the specific changes or additions to the standard methods that are required for their application to the MS2 system.

We will denote the observable spot intensity as $I$, such that the emission state in frame $k$ is $I_k$. The underlying gene state, represented by $g_k$, is quantized into discrete levels of promoter activity. From one frame to the next, the probability of the hidden state switching from state $i$ to state $j$ is encoded by the transition matrix, $T_{ij}$, where $T_{ij} = P(g_{k+1} = j|g_k = i)$. The observable emissions are related probabilistically to the hidden state using continuous distributions: $E_i(x) = P(I_k = x|g_k = i)$. Often a single Gaussian distribution or a superposition of Gaussians is used as each emission distribution; for the MS2 system, the properties of the Gaussian can be calculated from the initiation rate using **Equations 3.31** and **3.32**.

The utility of hidden Markov models relies on the assumptions that the probability of observing the emission state $I_k$ and the probability of switching to a state in frame $(k + 1)$ depend only on the current state $g_k$. While introducing some restrictions, these assumptions allow the probability of observing a long sequence of observations to be calculated efficiently (or indeed within astronomical timescales). By defining the probability $P(g_{k+1} = j|I_{k+1} = x, g_k = i) = T_{ij}E_j(x)$, it is possible to calculate the conditional probability of an observed sequence $P(data|\Theta)$ given the system parameters $\Theta = \{T_{ij}, E_i(x)\}$ in a single pass through the sequence, using the so-called forward algorithm. A similar definition of the conditional probabilities moving backwards through the sequence allows the likelihood of an individual underlying state $P(g_i|data, \Theta)$ to be calculated (the forward-backward algorithm). The suitability of a given model parameter set $\Theta$ is describing the data can be assessed using $P(data|\Theta)$, referred to as the likelihood, which is calculated using the forward algorithm. Therefore, an aim of using this approach is to find the parameter set with the maximum likelihood. This can be achieved using the Baum-Welch algorithm, an expectation maximisation (EM) method which iteratively converges towards a maximum of the likelihood in parameter space.

### 4.2 Modifications for the MS2 system

The clearest issue with a Hidden Markov approach in modelling the MS2 system of live transcription is the finite retention time of polymerases at the site of transcription. Not only do polymerases make different contributions to the intensity depending on their location on the gene, but the processivity of polymerases means that the intensity depends on the rate of initiation in the previous 2-3 min, rather than depending solely on the current promoter state, violating the Markov assumption. One approach to incorporate the history of the hidden state sequence is to add further hidden states encoding the promoter state in the previous $n$ frames, as far back as required. For the so-called nth-order model, the hidden state configuration is $\{g_{0,k}, g_{-1,k}, g_{-2,k} \ldots\}$, and the transition matrix is then modified so that $g_{0,k+1}$ is sampled from $T_{ij}$ using $g_{0,k}$, and the higher order history states are deterministically passed from the previous frame: $g_{-1,k+1} = g_{0,k}$, $g_{-2,k+1} = g_{-1,k}$, etc.

For the MS2 system, it is not the promoter state per se that determines the spot intensity, but rather the number of polymerases initiated, therefore we construct a hidden state architecture encoding the current promoter state and the number of polymerases initiated in previous frames that are still attached and therefore contribute to the intensity: $\{g_k, p_{0,k}, p_{-1,k}, p_{-2,k}, p_{-2,k}\}$. The dimensionality of the hidden states required depends on the retention time of a polymerase's RNA at the TS and the frame interval of imaging. The quantities $p_{m,k}$ effectively represent the number of polymerases present on different segments of the gene. Transitions from one frame to the next are determined by a set of transition matrices, with the promoter state randomly switching as before, the number of polymerases in the first segment being sampled from a Poisson variable of the current promoter state, and subsequent polymerase segments passed processively from the previous segment in the previous frame:

$$P\left(g_{(k+1)} = j | g_k = i\right) = G_{i,j} \tag{4.1}$$

$$p_{0,k} = \mathrm{Pois}\left(t_{frame} * \lambda(g_k)\right) \tag{4.2}$$

$$p_{m,k+1} = p_{(m-1),k} \ \text{ for } m = -1, -2, \dots \tag{4.3}$$

If the polymerases are not processive, that is, there is a non-zero probability of premature termination and cleavage, *Equation 4.3* may be modified appropriately to reflect the probabilistic passing of polymerases from one segment to the next.

The segments are related to the physical location on the gene, and so polymerases in different segments will contribute a different number of loops to the overall intensity. In this way, the emission state is constructed by summing the number of polymerases in each segment, multiplied by the number of loops per polymerase, $c_{-m}$, for that segment, combined with additive random measurement noise:

$$I_k = N\left(\mu = \sum_{m=0}^{M} c_{-m} p_{m,k}, \sigma^2 = \sigma_{noise}^2\right)$$

In practice the uncertainty arising from the variation in loop number across a segment is smaller than the measurement noise and can be subsumed into this.

## 4.3 Training models on data using the Baum-Welch algorithm

The complexity of the hidden state architecture for the polymerase hidden Markov model (pHMM) means that the vast majority of transitions from one hidden state to another have zero probability. Therefore, we developed a custom versions of the forward-backward and Baum-Welch algorithms using sparse matrix methods. During Baum-Welch optimization, the emission states are held fixed rather than being recalculated in each iteration, to prevent overfitting of the data and convergence to non-physical values for the measurement noise $\sigma_{noise}$ and segment loop number $c_m$. Therefore the measurement error and segment loop number must be known *a priori* and cannot be inferred using the expectation maximization (EM) of the Baum-Welch method. Furthermore, at each iteration we impose the constraint of Poisson activity within each promoter state, effectively calculating the best-fit value of the initiation rate of each hidden state, $\lambda(g_k)$. As shown by the results of the forward-backward algorithm (main text *Figure 1E*) and Gibbs sampling (see below), this does not prevent non-Poisson behaviour being observed when a model of long-lived discrete states is not appropriate.

The inputs to our Baum-Welch algorithm are the system parameters, $\sigma_{noise}$, calculated from autocorrelation measurements, the coefficients $c_m$ calculated from FRAP experiments, initial guesses for the initiation rate for each promoter state and the rates of transition between these states, and experimental data calibrated in terms of the number of loops at the transcription site. The algorithm produces estimates for the most likely values for the initiation and transition rates, and the likelihood $P(data|\Theta)$, which can be used to compare different hidden state architectures. The algorithm is run until the relative change in each of the parameters in $\Theta$ is smaller than a tolerance of $10^{-5}$, up to a maximum of 500 iterations.

The HMM framework was tested on simulated data of varying numbers of states, of known initiation and transition rates. As starting values, physically feasible values matching the spread in initiation rates and timescale of variability were supplied to the Baum-Welch algorithm, and the most likely system parameters were compared with the ground truth values of number of states, initiation rates and transition rates which are known as inputs to the simulations. The accuracy of the initial estimates influences the number of iterations required for convergence, but for physically realistic values did not affect the final convergence.

## 4.4 Gibbs sampling of polymerase sequences

To estimate the underlying initiation state and number of new polymerases in each frame of each cell, a Gibbs sampling algorithm was designed, taking account of the fact that a change in a polymerase sample affects the likelihood of multiple frames. Briefly, cells were considered one at a time. For each cell, the hidden states are initialized at random by sampling the next state based on *P(next|prev, intensity)*. A time point is chosen at random, and the hidden state for this time point is resampled using the probability of each hidden state *P(H(k)|I(k),H(k-1),H(k+1))*, which takes account of the spot intensity, the transition from the previous frame and the transition to the next frame. This resampling procedure was repeated many times before recording a sample of the hidden state sequence. The interval between taking samples of the hidden state sequence was defined such that each time point is resampled 15 times on average between each recorded sample. The distribution of polymerases initiated in each state calculated using Gibbs sampling were consistent with the data obtained using the forward-backward algorithm.

## 4.5 Optimizing model complexity

The utility of a particular model can be assessed by comparing simulations using the model to experiments. If the ground truth model has been identified then simulations of the model should fully reproduce the experimental data. Analysis of the hidden Markov fits for simple models with small numbers of discrete states indicates such models do not sufficiently account for the complexity of the data (*Figure 1E*). In order to distinguish between competing models we looked into standard methods of balancing goodness of fit and model complexity. Those most widely used are the Akaike Information Criterion (AIC) (*Singer et al., 2014*) and Bayesian Information Criterion (BIC), which both penalize the number of free parameters in the model, albeit with different weights (*Aho et al., 2014*). The AIC is a standard approach for finding the most accurate model when the real situation is likely to be highly complex (*Burnham and Anderson, 2002*). The BIC is most effective for simple systems and assumes the ground truth model is one of the candidate models to be tested. Since our data was more complex than any of the models we fit, the AIC was deemed the most appropriate for our study.

## Appendix 5 Additional details

### 5.1 Negative binomial distribution of cytoplasmic RNA number

For a gene with a constant Poisson rate of mRNA production, the number of mRNA molecules in the cytoplasm is a balance between the rates of production and decay. When the mRNA lifetime is a simple exponential distribution (one rate-limiting step for degradation) with lifetime $\tau_{RNA}$, the number of cytoplasmic mRNAs is a well-characterized Poisson distribution. The effect of transcriptional bursting is to broaden the distribution of cytoplasmic mRNA number, increasing the ratio of the variance to the mean. In this situation, the distribution of mRNA per cell, $x$, has been shown to be characterized by a negative binomial (NB) distribution (**Paulsson et al., 2000**).

$$P(x,r,p) = \binom{x+r-1}{x} p^r (1-p)^n$$

The Poisson distribution is a limiting case of the negative binomial distribution as $r \rightarrow \infty$. In the context of transcriptional bursting, the NB parameters $r$ and $p$ can be related to bursting characteristics, with the frequency of bursts given by $r\tau_{RNA}$ and the average number of mRNA produced per burst by $(1-p)/p$.

The parameters $r$ and $p$ and their confidence intervals were calculated from simulated smFISH distributions using the MATLAB function nbinfit.

### 5.2 Effect of quantizing pulse duration

$$P(x|X) = G(x) = \begin{cases} 0 & \text{if } x < X-1; \\ 1-|x-X| & \text{if } X-1 < x < X+1; \\ 0 & \text{if } x > X+1; \end{cases}$$

$$P(X=k) = \int_{x=k-1}^{x=k+1} G(x)P(x)dx \tag{5.1}$$

$$= \int_{x=k-1}^{x=k} (1-k+x)\Lambda e^{-\Lambda x}dx + \int_{x=k}^{x=k+1} (1-x+k)\Lambda e^{-\Lambda x}dx \tag{5.2}$$

$$= \int_{k-1}^{k} x\Lambda e^{-\Lambda x}dx - \int_{k}^{k+1} x\Lambda e^{-\Lambda x}dx + (1-k)\int_{k-1}^{k} \Lambda e^{-\Lambda x}dx$$
$$+ (1+k)\int_{k}^{k+1} \Lambda e^{-\Lambda x}dx \tag{5.3}$$

$$= (k-1)e^{-\Lambda(k-1)} - ke^{-\Lambda k} + \frac{1}{\Lambda}e^{-\Lambda(k-1)} - \frac{1}{\Lambda}e^{-\Lambda k} + (k+1)e^{-\Lambda(k+1)} - ke^{-\Lambda k}$$
$$+ \frac{1}{\Lambda}e^{-\Lambda(k+1)} - \frac{1}{\Lambda}e^{-\Lambda k} + (k-1)e^{\Lambda k} - (k-1)e^{-\Lambda(k-1)}$$
$$+ (k+1)e^{-\Lambda k} - (k+1)e^{-\Lambda(k+1)} \tag{5.4}$$

$$= e^{-\Lambda(k-1)}\left(k-1+\frac{1}{\Lambda}-(k-1)\right)$$

$$+ e^{-\Lambda k}\left(-k-\frac{1}{\Lambda}-k-\frac{1}{\Lambda}+(k-1)+(k+1)\right) \tag{5.5}$$

$$+ e^{-\Lambda(k+1)}\left(k+1+\frac{1}{\Lambda}-(k+1)\right)$$

$$= e^{-\Lambda(k)}\frac{(e^\Lambda - 2 + e^{-\Lambda})}{\Lambda}$$

$$P(X=k) = e^{-\Lambda k}\frac{4}{\Lambda}\sinh^2(\Lambda/2) \tag{5.6}$$

$$P(X=k) = e^{-k/\lambda}4\lambda\sinh^2\left(\frac{1}{2\lambda}\right)$$

$$\Rightarrow P(X=k)\,\alpha\,e^{-k/\lambda}$$

## 5.3 Measurement of the decay time

### 5.3.1 Normalization constant

$$P(x) = Ae^{-x/\lambda} \tag{5.7}$$

$$1 = A\sum_{x=1}^{\infty}e^{-x/\lambda} \tag{5.8}$$

$$\frac{1}{A} = e^{-1/\lambda} + e^{-2/\lambda} + e^{-3/\lambda} + \ldots \tag{5.9}$$

$$= \frac{e^{-1/\lambda}}{1-e^{-1/\lambda}} \tag{5.10}$$

$$= \frac{1}{e^{1/\lambda}-1} \tag{5.11}$$

$$A = e^{1/\lambda} - 1 \tag{5.12}$$

### 5.3.2 Calculation of mean

$$\langle X \rangle = \sum_{x=1}^{\infty}xP(x) \tag{5.13}$$

$$= (e^{1/\lambda}-1)\sum_{x=1}^{\infty}xe^{-x/\lambda} \tag{5.14}$$

$$\text{Let } \Lambda = 1/\lambda \tag{5.15}$$

$$= (e^\Lambda - 1) \sum_{x=1}^{\infty} x e^{-\Lambda x} \tag{5.16}$$

$$= (e^\Lambda - 1) \sum_{x=1}^{\infty} -\frac{\partial}{\partial \Lambda} (e^{-\Lambda x}) \tag{5.17}$$

$$= -(e^\Lambda - 1) \frac{\partial}{\partial \Lambda} \sum_{x=1}^{\infty} e^{-\Lambda x} \tag{5.18}$$

$$= -(e^\Lambda - 1) \frac{\partial}{\partial \Lambda} \left( \frac{1}{e^\Lambda - 1} \right) \tag{5.19}$$

$$= -(e^\Lambda - 1) \left( -\frac{e^\Lambda}{(e^\Lambda - 1)^2} \right) \tag{5.20}$$

$$\langle X \rangle = \frac{e^{1/\lambda}}{e^{1/\lambda} - 1} \tag{5.21}$$

$$\Rightarrow \lambda = \frac{1}{\ln \left( \frac{\langle X \rangle}{\langle X \rangle - 1} \right)} \tag{5.22}$$

### 5.3.3 Approximations for small and large $\lambda$

$$\text{as } \lambda \longrightarrow 0, \quad \langle X \rangle \longrightarrow 1 \tag{5.23}$$

$$\text{as } \lambda \longrightarrow \infty, \quad \langle X \rangle = e^\Lambda (e^\Lambda - 1)^{-1} \tag{5.24}$$

$$\longrightarrow \left( 1 + \Lambda + O(\Lambda^2) \right) \left( \Lambda + \frac{\Lambda^2}{2} + O(\Lambda^3) \right)^{-1} \tag{5.25}$$

$$\longrightarrow \frac{\left( 1 + \Lambda + O(\Lambda^2) \right)}{\Lambda} \left( 1 - \frac{\Lambda}{2} \right) \tag{5.26}$$

$$\longrightarrow \frac{1}{\Lambda} \left( 1 + \Lambda(1 - 1/2) + O(\Lambda^2) \right) \tag{5.27}$$

$$\longrightarrow \frac{1}{\Lambda} + 1/2 + O(\Lambda) \tag{5.28}$$

$$\langle X \rangle \longrightarrow \lambda + 1/2 \tag{5.29}$$

