## [Decision Letter]

Thank you for submitting your work entitled "A Continuum Model of Transcriptional Bursting" for consideration by *eLife*. Your article has been reviewed by two peer reviewers, and the evaluation has been overseen by Robert Singer (Reviewing Editor) and Detlef Weigel (Senior Editor). One of the two reviewers, Arjun Raj, has agreed to share his identity.

The reviewers have discussed the reviews with one another and the Reviewing Editor has drafted this decision to help you prepare a revised submission.

Summary:

There was enthusiasm for the analysis of the data, but some concern that only a single gene was characterized. In addition both reviewers felt that the text needs to be clarified to have to broader significance be evident and that earlier work should be dealt with more extensively. In the ensuing discussion, both reviewers agreed that the work is important and should be published with their suggested revisions.

Reviewer 1 agreed that "it is probably too much to ask to get more genes, and that the strength of their work lies in the impressive depth of their analysis – and I would be okay if they better discuss the limitations of their model system: one gene that might have its peculiarities. Concerning general conclusions about the prevalence of burstiness: I would be cautious here since this is a highly active housekeeping gene, and it is known that some constitutive genes behave in a non-bursty manner; also to me this is almost semantic, since the difference between getting from very low initiation rate to high initiation rate (no bursting) or from null initiation rate to high initiation rate (bursting) is to my opinion not deeply significant. I fully agree with you concerning the writing, the text needs thorough reformulation to drastically improve its clarity, especially the take home messages should be stated in a more down-to-earth manner to give some meat to non-specialists, and interesting findings such as the time scales of changes in initiation rates should be better discussed."

Arjun Raj commented: “I think [Reviewer 2's] points are spot on. I think that the authors should carefully discuss the fact that the gene they studied may not be fully representative, in particular because it is a housekeeping gene. I agree that as stated, the difference seems semantic between bursting and non-bursting, and the authors should more clearly explain how exactly their results differ in that regard.”

We ask for the following essential revisions:

1) Improve the clarity of the text so as to make it understandable to a wider audience.

2) Provide a more extensive discussion of previous work.

3) Include caveats as to the generalizations from analysis of a single gene (unless you can add analyses of additional genes).

Minor comments are provided in the reviews below.

*Reviewer #1:* In this paper, the authors critically evaluate models for transcriptional bursting in light of their time-lapse experimental data. They delve deeply into the data generated by their MS2-GFP system on a particular gene, finding through simulations that one can explain the periods of apparent transcriptional activity and inactivity even with just a one-state model with no bursting at all. Through further analysis, they show that a model in which transcriptional activity consists of a continuum of activity states fits their data the best.

I think this paper is really important. In the field of single cell transcriptional dynamics, the notion of bursts has been the dogma for some time now, but I don't believe anyone has really put the many assumptions to this degree of rigorous testing before. I think this paper raises the possibility that the prevalence of transcriptional bursts may actually be considerably less than believed, and this work will raise the bar for analysis in this field.

My primary concern with the manuscript is that, as written, it is essentially incomprehensible to those outside the field, and frankly still fairly difficult to understand for those in the field. Some of this is certainly due to the technical nature of the work, but I think the authors could really do a much better job of making the main points accessible to a broader audience. Furthermore, beyond accessibility to a broad audience, there are several places where the reader is left guessing as to important details and definitions in the model and conclusions.

That said, I still think this is a really nice paper that will have an impact.

Here are some specific notes for improvement:

– As mentioned, the writing is often dense, with various things undefined and thus hard to understand. Here are a few examples:

"… loop contribution of polymerases along the gene": what is a "loop contribution"?

The description of model in "pulse theory" section is very terse and not particularly clear. Given that this is crucial for the paper, I recommend rewriting to make clearer. Currently, one has to read the supplement to make any sense out of this section, which I think is not great for clarity. Can the authors give some intuition for why taking all these factors into account could lead one to see something that looks exponential? There must be a simple intuition for this. Is it simply a detection threshold?

It is important to be very clear exactly what the definitions are for initiation, elongation, transcriptional burst parameters and their relationships in the model.

"… shows that the length of time for which a spot is visible is not trivially related to switches between proposed gene states in two state models…": What is the relationship?

"As depicted in Figure 2, the model of regulation depends strongly on the number of polymerases which contribute to the pulses and intervals." I found this statement confusing-precisely what models and what is the dependency?

– I think it's really important that the authors mention that they are just looking at a single gene, and parameters (and thus conclusions) may vary dramatically from gene to gene. For instance, how would their results mesh with the recent Molina et al. data (10.1073/pnas.1312310110)? I think that showed pretty convincing bursts, and should be well above background. Can the authors discuss in the context of their model?

In line with the authors' conclusions, it may be worth discussing that in Padovan-Merhar Mol Cell 2015, we found that once volume gets factored in, some genes (e.g., GAPDH) actually are statistically indistinguishable from Poisson (although it is harder to experimentally rule *out* bursting there). So perhaps a reinterpretation of previous bursting results is warranted as well. Padovan-Merhar also showed extrinsic effects on burst size, perhaps worth mentioning that these extrinsic effects may also play a role (and are not accounted for in the model currently, I think).

– Can the authors more intuitively describe their penalty for model complexity? The result that the ever-increasing goodness of fit indicates a continuum of states would seem to depend heavily on the particular penalty-if one adopted a stronger penalty, then wouldn't the number of states eventually stop increasing?

– The figures in general are somewhat confusing and confusingly labeled. E.g., it's not a priori obvious what the "loops" in Figure 1 are. I think it would really help to take a fresh look and try to improve the presentation.

– Others have found that transcriptional regulation can affect either burst size or burst frequency/fraction. Can the authors discuss the implications their models have for these results?

*Reviewer #2:*

Corrigan et al. analyze transcriptional kinetics of the actin5 gene in Dictyostelium, using the MS2 system to monitor transcription by live fluorescence imaging. Transcriptional bursting has traditionally been described using models defining genes as being either on or off. Here the authors show that the on-state of gene activity is not well described by a single state, but rather by a continuum of states with varying initiation rates. The authors perform thorough analysis of a large dataset, with careful determination of their detection threshold and the potential effect of varying elongation rates, and show that the actin5 gene is characterized by varying initiation rates, and the access to the highest rates is prevented by mutations in the TATA box. There are no obvious issues with the technical quality of the data or simulation experiments, even though the model system used with the reporter present in two copies (since cells are in G2) slightly complicates this type of analysis. However, there are some issues with the novelty and the scope of their findings:

Major comments:

– The authors overstate the novelty of their results as some earlier work, although cited, is not satisfactorily discussed in the light of the present findings. The same group (Corrigan et al., Current Biology 2014) and others (Molina et al., PNAS 2014) has shown that transcription rates of on-activity windows can vary significantly over time for genes responding to extracellular signalling. Since the fact that a gene can exhibit different rates of transcription within its periods of activity was shown before, the novelty of this work lies mainly in the extension of these findings to demonstrate that varying transcription rates i) can occur in a "steady state" situation and ii) arise mainly from variations in the transcription initiation rates.

– This work is based on the study of a single gene, from which the authors drive general conclusions about the existence of a continuum of on-states for gene expression. Although it seems reasonable that such fundamental processes will hold true for a number of genes, the data presented here offers limited insights on the molecular mechanisms at their origin or on their potential biological consequences. Having several genes to assess the variability in the magnitude of these initiation rate fluctuations, as well as the time scales allowing to switch from one state to another would be of major interest and shed light on the biological significance of these findings.

---

## [Author Response]

*We ask for the following essential revisions:*

*1) Improve the clarity of the text so as to make it understandable to a wider audience.*

The manuscript has been extensively rewritten, with specific emphasis on making the more technical issues intuitive to the general reader, but retaining sufficient information on the technical aspects for the more quantitative audience. In particular, there has been a substantial reworking of the theory and probabilistic modeling sections, in addition to a general consideration of all the text. We have assessed our improvements by getting three cell biologists and one physicist with backgrounds in gene regulation to check for readability and scientific content.

*2) Provide a more extensive discussion of previous work.*

We have incorporated more discussion of these into the Introduction, where recent papers by Darzacq, Naef and Raj labs have been used to highlight the need for a more thorough analysis of the popular one and two state models. We have also reassessed these papers in the light of our data in the concluding paragraph, indeed highlighting these papers as suggestions that our work may have great relevance for genes with more complex regulation, such as in mammalian cells.

*3) Include caveats as to the generalizations from analysis of a single gene (unless you can add analyses of additional genes)*

We include a direct statement in the Results section, and discussion of this feature of our study as the entry to the concluding paragraph of the paper.

*Minor comments*

Reviewer #1:

*"*… *loop contribution of polymerases along the gene": what is a "loop contribution"?*

*The description of model in "pulse theory" section is very terse and not particularly clear. Given that this is crucial for the paper, I recommend rewriting to make more clear. Currently, one has to read the supplement to make any sense out of this section, which I think is not great for clarity. Can the authors give some intuition for why taking all these factors into account could lead one to see something that looks exponential? There must be a simple intuition for this. Is it simply a detection threshold?*

We have addressed these comments, and general readability with a substantial rewrite of this section of the paper (see above). We describe intuitively how initiation, elongation, detection threshold and imaging frame contribute to pulse duration.

*It is important to be very clear exactly what the definitions are for initiation, elongation, transcriptional burst parameters and their relationships in the model.*

These are now clearly defined in the text as we first introduce the simulations and theory section

*"*… *shows that the length of time for which a spot is visible is not trivially related to switches between proposed gene states in two state models*…*": What is the relationship?*

We have reworded the sentence to highlight the additional contributions to pulse duration:

“Overall, this theory shows that the length of time for which a spot is visible is not simply related to switches between proposed gene states in two state models and explains the dependence of the pulse duration on the imaging signal-to-noise and frame interval.”

*"As depicted in Figure 2, the model of regulation depends strongly on the number of polymerases which contribute to the pulses and intervals." I found this statement confusing-precisely what models and what is the dependency?*

This is indeed unclear – we have provided a more gentle description of the problem of model selection without quantitative data, which relates to the figure.

*– I think it's really important that the authors mention that they are just looking at a single gene, and parameters (and thus conclusions) may vary dramatically from gene to gene. For instance, how would their results mesh with the recent Molina et al. data (10.1073/pnas.1312310110)? I think that showed pretty convincing bursts, and should be well above background. Can the authors discuss in the context of their model?*

*In line with the authors' conclusions, it may be worth discussing that in Padovan-Merhar Mol Cell 2015, we found that once volume gets factored in, some genes (e.g., GAPDH) actually are statistically indistinguishable from Poisson (although it is harder to experimentally rule* out *bursting there). So perhaps a reinterpretation of previous bursting results is warranted as well. Padovan-Merhar also showed extrinsic effects on burst size, perhaps worth mentioning that these extrinsic effects may also play a role (and are not accounted for in the model currently, I think).*

We agree with all these points – there is now a strong section in both the Introduction and Discussion which more fully integrates the recent work by Raj, and the Darzacq and Naef groups with our own.

*– Can the authors more intuitively describe their penalty for model complexity? The result that the ever-increasing goodness of fit indicates a continuum of states would seem to depend heavily on the particular penalty-if one adopted a stronger penalty, then wouldn't the number of states eventually stop increasing?*

This section has been reworded extensively, to read more intuitively to cell biologists. The AIC is the most effective measure for highly complex datasets. We have added a section into the hidden Markov section of the supplementary material discussing this choice of measure.

*– The figures in general are somewhat confusing and confusingly labeled. E.g., it's not a priori obvious what the "loops" in Figure 1 are. I think it would really help to take a fresh look and try to improve the presentation.*

The figures have been extensively relabeled and we have provided a simple cartoon as Figure 1 so the reader can get a sense of the key aspects of the model. The figures for running of the hidden Markov model on the simulated data (Figure 3) have been replaced with figures we more conventionally use for presentations, which are more intuitive and in keeping with the other figures. To make more of the initiation rate fluctuations (as requested by Reviewer 2), we have included an additional Figure 3 with associated text highlighting the timescale.

*– Others have found that transcriptional regulation can affect either burst size or burst frequency/fraction. Can the authors discuss the implications their models have for these results?*

As mentioned above, these studies have now been more explicitly discussed in the Introduction and Discussion.

Reviewer #2:

*– The authors overstate the novelty of their results as some earlier work, although cited, is not satisfactorily discussed in the light of the present findings. The same group (Corrigan et al., Current Biology 2014) and others (Molina et al., PNAS 2014) has shown that transcription rates of on-activity windows can vary significantly over time for genes responding to extracellular signalling. Since the fact that a gene can exhibit different rates of transcription within its periods of activity was shown before, the novelty of this work lies mainly in the extension of these findings to demonstrate that varying transcription rates i) can occur in a "steady state" situation and ii) arise mainly from variations in the transcription initiation rates.*

We have been more explicit about the earlier work. The Molina paper finds different parameters in response to different added signals, calculating production rates for populations of cells exposed to distinct stimuli. It is very interesting, but here we directly quantify transcriptional dynamics rather than infer them from protein. Our work examines the heterogeneous spectrum of single cell behaviours within a population, and finds that cells do not have fixed bursting parameters. We have added a sentence on the steady state nature of the experiment.

*– This work is based on the study of a single gene, from which the authors drive general conclusions about the existence of a continuum of on-states for gene expression. Although it seems reasonable that such fundamental processes will hold true for a number of genes, the data presented here offers limited insights on the molecular mechanisms at their origin or on their potential biological consequences. Having several genes to assess the variability in the magnitude of these initiation rate fluctuations, as well as the time scales allowing to switch from one state to another would be of major interest and shed light on the biological significance of these findings.*

As described above, we have highlighted the potential caveats of looking at a single gene, although we make a reasonable speculation that the high level of complexity found here for a relatively “simple” system may yet underrepresent the complexity of a mammalian gene with multiple long range enhancers.